# Nanoscale 3D spatial addressing and valence control of quantum dots using wireframe DNA origami

Chi Chen [1], Xingfei Wei [2], Molly F. Parsons [1], Jiajia Guo [3,6], James L. Banal[1,7], Yinong Zhao [4], Madelyn N. Scott [3], Gabriela S. Schlau-Cohen [3], Rigoberto Hernandez [2,4,5] & Mark Bathe [1] ✉

Control over the copy number and nanoscale positioning of quantum dots (QDs) is critical to their application to functional nanomaterials design. However, the multiple non-specific binding sites intrinsic to the surface of QDs have prevented their fabrication into multi-QD assemblies with programmed spatial positions. To overcome this challenge, we developed a general synthetic framework to selectively attach spatially addressable QDs on 3D wireframe DNA origami scaffolds using interfacial control of the QD surface. Using optical spectroscopy and molecular dynamics simulation, we investigated the fabrication of monovalent QDs of different sizes using chimeric single-stranded DNA to control QD surface chemistry. By understanding the relationship between chimeric single-stranded DNA length and QD size, we integrated single QDs into wireframe DNA origami objects and visualized the resulting QD-DNA assemblies using electron microscopy. Using these advances, we demonstrated the ability to program arbitrary 3D spatial relationships between QDs and dyes on DNA origami objects by fabricating energy-transfer circuits and colloidal molecules. Our design and fabrication approach enables the geometric control and spatial addressing of QDs together with the integration of other materials including dyes to fabricate hybrid materials for functional nanoscale photonic devices.

Quantum dots (QDs) are colloidally stable, inorganic semiconductor nanoparticles that exhibit quantum confinement[1]. Because of their tunable photophysics and excellent photostability compared with organic dyes, QDs have been used as bulk active materials in light-emitting diodes[2,3], photovoltaic solar cells[4], and photon detectors[5]. Beyond these applications, control over the spatial relationships of individual QDs on the 1–100 nm length-scale is key to applying them as

single-photon sources[6,7], biological sensors and imaging probes[8–10], and processing units for non-classical computing[11]. Realizing functional QDs for these applications requires discrete valence control over materials attached to the QD surface including dyes, proteins, and nucleic acids. However, controlling the valence of such materials on QDs remains an outstanding challenge because numerous non-specific reactive sites intrinsic to the semiconductor surface of QDs, in general,

[1]Department of Biological Engineering, Massachusetts Institute of Technology, Cambridge, MA 02139, USA. [2]Department of Chemistry, Johns Hopkins University, Baltimore, MD 21218, USA. [3]Department of Chemistry, Massachusetts Institute of Technology, Cambridge, MA 02139, USA. [4]Department of Chemical & Biomolecular Engineering, Johns Hopkins University, Baltimore, MD 21218, USA. [5]Department of Materials Science and Engineering, Johns Hopkins University, Baltimore, MD 21218, USA. [6]Present address: Bionic Sensing and Intelligence Center, Institute of Biomedical and Health Engineering, Shenzhen Institutes of Advanced Technology, Chinese Academy of Sciences, Shenzhen 518055, China. [7]Present address: Cache DNA, Inc., 200 Lincoln Centre Drive, Foster City, CA 94404, USA. ✉e-mail: mark.bathe@mit.edu

leads to a statistical distribution of the desired monovalent product together with multivalent, as well as unreacted, byproducts[12,13]. The presence of multivalent QD byproducts can lead to undesired oligomerization reactions, which may impede the fabrication of discrete and well-defined QD superstructures or QD assemblies involving other materials[14].

To address the challenge of controlling the valence of materials attached to QD surfaces, various strategies have been developed. These include principally covalent or non-covalent attachment of monovalent streptavidin[15] and surface adsorption of chimeric DNA strands composed of a contiguous sequence tract with a phosphorothioate (ps) backbone followed by a sequence tract with a phosphate (po) backbone (Supplementary Table 1)[16–18]. In one application of this latter approach, Kelley et al. reported the one-step synthesis of CdTe QDs with one to five single-stranded DNA (ssDNA) binding sites[16]. However, this one-pot approach required aqueous synthesis of QDs, which can often lead to the formation of electron or hole traps inside the interior of the lattice due to poor QD crystallinity[19] and reduce fluorescence quantum yield. In a related strategy using a ps-backbone, Jun, Gartner et al. developed a ligand-exchange strategy to render the chimeric ssDNA strand wrapping strategy compatible with commercially available QDs functionalized with organic ligands[17]. Fan et al. further developed this strategy using different chimeric DNA complexes per QD to program QD valence[18]. More recently, ssDNA containing variable poly(deoxyadenosine) domains that bind to gold nanoparticles (AuNPs) was used to program the valence and relative positions of AuNPs without precise 2D or 3D structural control[20]; yet no such a strategy for controlling QD valence with nanometer-scale 3D structural positioning exists.

DNA nanotechnology is a powerful approach to organizing secondary molecules ranging from small-molecule dyes to nanoparticles with quantitative control over copy number and nanometer-scale 3D spatial precision[21–23]. In particular, DNA origami[24] that employs a long ssDNA scaffold strand to fabricate 2D and 3D wireframe objects[25–28] offers the unique ability to control relative distances, angles, and copy numbers of secondary molecules in a nearly arbitrary manner, including asymmetric spatial organization[23,29]. In contrast to tile-based self-assembly approaches that typically suffer from limited yields[30–32], wireframe DNA origami typically offers an excellent yield (above 90%) of nearly arbitrary 2D and 3D target DNA-based nanostructures together with site-specific functionalization[33–35]. And the use of two-helix bundle (2HB) or six-helix bundle (6HB) edges in wireframe DNA origami designs also offers the unique combination of high structural fidelity, mechanical rigidity, quantitative fabrication yield, and nearly arbitrary geometric control on the 10–100 nm scale[25,26].

During the past decade, DNA nanotechnology has been used to organize individual QDs in 1D, 2D, and 3D systems[36–42]. For example, Liu et al. demonstrated the organization of QD–DNA conjugates by complementary base pairing to triangular- and rectangular-shaped DNA origami structures[36]. In another example, Liedl et al. fabricated "planet–satellite" nanoclusters with controlled sizes up to 500 nm using DNA-origami-guided self-assembly[37]. And Diaz et al. recently demonstrated the efficient conjugation of QDs to DNA nanostructures using QDs with a peptide nucleic acid (PNA) containing a hexahistidine peptide motif[40]. However, the preceding strategies still suffer from multiple ssDNAs and PNAs that are typically bound non-specifically to the QD surface, which increases the probability that a QD binds to more than one DNA nanostructure and consequently reduces the purity of target assemblies. Thus, controlling the ssDNA valence of individual QDs is crucial to avoiding multiple DNA origami objects from attaching to a single QD. An alternative strategy that used a single, flexible icosahedral wireframe DNA tile-based assembly to encapsulate single QDs resulted in monofunctionalized QDs[38]. However, this strategy is not easily generalized to libraries of QD-DNA assemblies of various sizes and geometries, and the intrinsic flexibility

of the single duplex objects employed results in low shape fidelity compared with 2HB and 6HB designs[25–27]. This may in turn negatively impact functional materials applications in photonics[43], excitonics[44,45], and molecular and cell biophysics[46]. Moreover, this non-specific encapsulation approach typically leads to relatively low yield of the final product[38]. As yet another alternative, Gang et al. demonstrated a more general material voxel design strategy that involved assembling 3D lattices of wireframe DNA origami structures that individually contained metallic nanoparticles, semiconductor nanoparticles, or proteins[41]. However, they employed bulky biotin–streptavidin that separates the QD surface and wireframe DNA origami object by at least 5 nm, impeding the implementation of efficient Förster resonant energy transfer (FRET)-based QD probes or FRET networks for biosensing and imaging, molecular logic and computing[47–52], as well as high resolution structural biology[53]. Thus, a facile and general yet robust strategy to immobilize QDs in arbitrary 3D geometric arrangements with controlled valence and high-resolution spatial positioning is still needed to achieve the fabrication of functional multivalent QD-DNA assemblies.

Toward this end, here we sought to overcome the preceding limitations by programming ssDNA valence on QD surfaces with wireframe DNA origami and chimeric ssDNA wrapping. Our synthetic approach offers control over valences of QD with tunable size and spectra, as well as relative 3D spatial positions of QDs and other materials such as dyes, which we term "valence-geocoded" QD (Fig. 1a). Specifically, we initially investigated monovalent chimeric ssDNA wrapping of QDs to control site-specific immobilization in a single wireframe DNA origami object. Target wireframe DNA origami objects were designed using the software ATHENA[29] to generate the scaffold and staple sequences required for self-assembly. Site-specific staple sequences contained an overhang with ps segment of varying length to wrap QDs of different sizes, or an overhang with a po segment to hybridize other dyes or immobilize QDs at arbitrary spatial positions on the wireframe DNA origami scaffold (Fig. 1b). To guide the design of QD-DNA assemblies, the relationships between ps-backbone length on chimeric ssDNA, QD size, and wireframe DNA origami object geometry were first investigated using experiment and computation. Our approach enabled the fabrication of diverse target assemblies including QD-based FRET networks and QD-based colloidal molecules (Fig. 1c) using chimeric ssDNA to maintain close proximity of the QD surface to the wireframe DNA origami conjugation sites. Our work offers a general framework to leverage highly programmable wireframe DNA origami to achieve geocoded colloidal assemblies with combinations of QDs and dyes.

## Results

### QD size and phosphorothioate (ps) tract length control valence and hybridization efficiency

To fabricate monovalent ssDNA-QDs via wrapping, commercial CdSe/ZnS core/shell QDs were first transferred from organic solvent to water using Zn-assisted phase transfer[17,18]. Ligand exchange was then used to replace a portion of 3-mercaptopropionic acid (MPA) with O-(2-mercaptoethyl)-O′-methyl-hexa (ethylene glycol) (mPEG) to obtain MPA/mPEG QDs. The addition of mPEG permitted the tuning of surface QD charge by varying mPEG concentration. Finally, ssDNAs containing ps tracts were incubated with the ligand-exchanged QD to prepare monofunctionalized QDs (Supplementary Fig. 1).

To generalize our approach of fabricating valence-geocoded QDs, we systematically investigated the impact of ps-backbone length and QD size on valence control and hybridization yield. 6, 8, and 14 nm QDs with emission wavelengths of 600 (QD600), 630 (QD630), and 660 nm (QD660), respectively, (Supplementary Figs. 2 and 3) were wrapped with ssDNA with ps-backbones with oligo(deoxyadenosine) tracts, denoted as A*. We used 5, 10, and 30 nucleotide (nt) A* to wrap QD600 and QD630, while 5, 30, and 50 nt A* tracts were used for

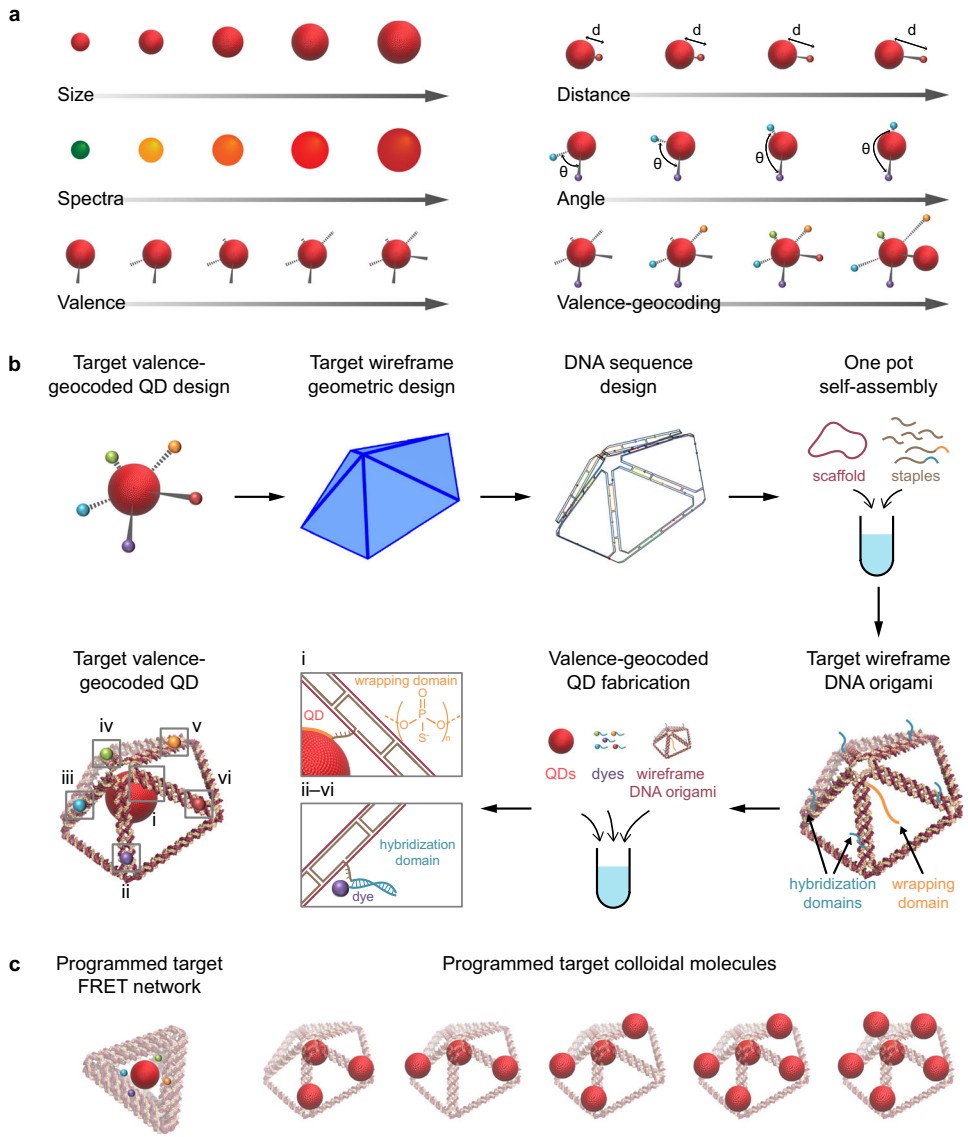

**Fig. 1 | Strategy to design and fabricate valence-geocoded QDs. a** Schematic of capabilities of valence-geocoded QD. We developed a strategy that combines the size and spectral tunability of QD with ssDNA wrapping strategies and wireframe DNA origami to spatially coordinate the 3D assemblies of QD networks.
**b** Schematic of workflow to design and fabricate target valence-geocoded QD via self-assembled wireframe DNA origami objects. The QD-wireframe DNA origami object binding site is labeled with i, while the valence binding sites are labeled with ii–vi. The ssDNA wrapping strategy and hybridization strategy can be applied to i–vi and ii–vi binding sites, respectively, according to the target valence-geocoded QD design. **c** Schematic of target assemblies, including programmed target FRET network and programmed target colloidal molecules.

QD660. All A* tracts also combined with a ssDNA sequence with po-backbone available for hybridization (Fig. 2a). In contrast, QDs were also conjugated with 3′-thiolated ssDNA, which normally resulted in binding multiple ssDNAs.

To validate valence control using the chimeric ssDNA strands, we developed a series of ratiometric spectroscopic assays by hybridization of complementary dye-labelled ssDNA to the po tract of the QD-wrapped chimeric ssDNA (Fig. 2a). The conjugated dye on the complementary ssDNA provides a distinct measurable signal in the absorbance and emission spectra of the QD-dye hybrids (Fig. 2b), which we used to quantify the ratio of QD-bound chimeric ssDNA to QD using absorbance spectrophotometry, emission spectrophotometry, and FRET efficiency with steady-state and time-resolved fluorescence measurements (Fig. 2c–e and Supplementary Figs. 4–6). The dye/QD molar ratio and FRET efficiency of the QD-dye hybrids were calculated using Eqs. (1–4) (Methods) according to the parameters from Fig. 2b and Supplementary Table 2. For QD600-AF647

and QD630-AF647 FRET pairs, QD donor photoluminescence (PL) intensities and lifetimes decreased with increasing ps-backbone length from 5 to 30 nt A* due to FRET (Supplementary Figs. 4 and 6). Comparison with theoretical estimates using Eq. (5) (Methods) with experimentally measured efficiencies showed donor-acceptor distances that increased from $5.1 \pm 0.2$ to $6.1 \pm 0.2$ nm (mean ± standard deviation; $n = 3$) for QD600-AF647 and QD630-AF647 FRET pairs, respectively (Fig. 2c, d). Using an agarose gel electrophoresis (AGE) assay, we observed that the electrophoretic mobility of QD-DNA hybrids relative to aqueous QD with neutral surface charge increased with increasing ps-backbone length (5, 10, and 30 nt A*) (Supplementary Fig. 7), indicating high purity of monovalent QDs[17,54]. The AGE assay also showed that the poor hybridization efficiencies observed using FRET measurements were not caused by bare QDs but by the presence of po domains that are inaccessible to hybridization on chimeric ssDNA with short A* tracts. Moreover, in the case of the QD-AF647 FRET in a distal configuration, FRET efficiencies

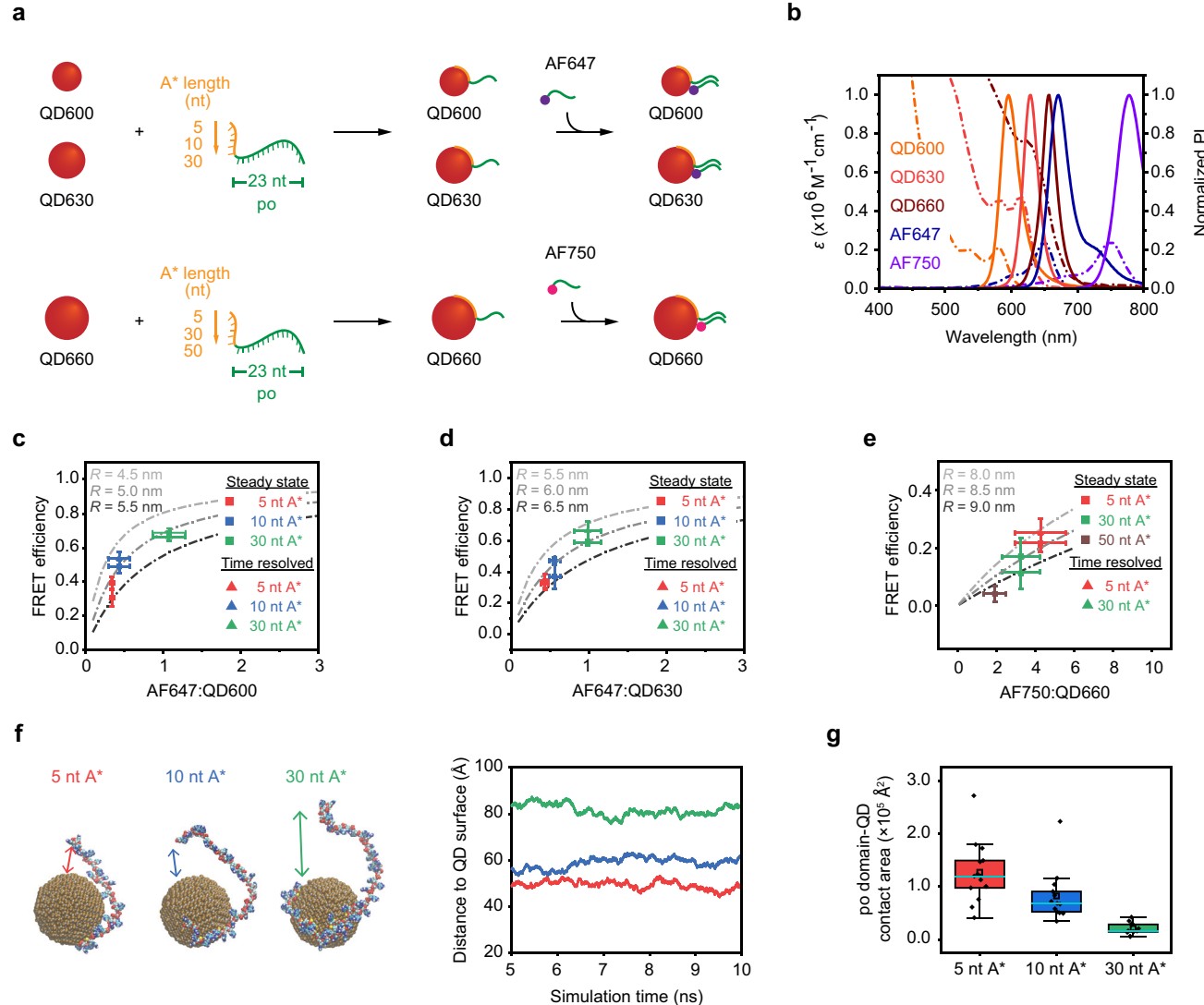

**Fig. 2 | ssDNA wrapping hypothesis. a** Schematic of ssDNA wrapping with chimeric ssDNA containing 5–50 nt A* and fixed po domain (23 nt) wrapping on QD600, QD630, and QD660, and QD valence validation using hybridization assay using AF647- or AF750-labelled complimentary strands. **b** Extinction coefficient (dash-dotted line) and photoluminescence (PL) spectra (solid line) of QD600, QD630, QD660, AF647 and AF750. **c**–**e** FRET efficiencies as a function of acceptors calculated theoretically (gray dash-dotted curves, Eq. (5)), and from QD emission intensities (Supplementary Fig. 4) and QD emission decay curves (Supplementary Fig. 6) of **c** QD600-AF647 and **d** QD630-AF647 and **e** QD660-AF750 FRET pairs (the number of AF647 or AF750 per QD were calculated from decomposed absorption spectra (Supplementary Fig. 5)). **f** Representative MD simulation structures of

ssDNA-wrapped 6 nm diameter QD and the distances of 5′-end of the 23 nt po-backbone to the QD surface for 5 nt A* (red), 10 nt A* (blue) and 30 nt A* (green). **g** Box-and-whisker plot of 23 nt po domain and QD contact area using MD simulation, hollow square and cyan solid line represent the mean value and median line, respectively. Error bars in **c**–**e** represent standard deviation of the mean ($n = 3$ replicates per group) for steady-state measurements and 95% confidence intervals for time-resolved measurements. Error bars in **g** represent the maximum and minimum values using 1.5 interquartile range (IQR) ($n = 13$ calculated from five different ensembles at different simulation times). Source data are provided as Source Data file.

were $10 \pm 2\%$ and $13 \pm 2\%$, and donor-acceptor distances increased to $8.3 \pm 0.3$ and $8.8 \pm 0.2$ nm (mean ± standard deviation; $n = 3$) for QD600-distal-AF647 and QD630-distal AF647 FRET pairs, respectively, which was consistent with the distance ranges calculated from potential QD-DNA duplex-dye geometries (Supplementary Fig. 8).

To further investigate monovalent functionalization of QDs using chimeric ssDNA wrapping, chimeric ssDNA with fixed total length (51 nt) but varying A* tract lengths (0–40 nt) and 5′-thiolated-ssDNA (51 nt) were incubated with QD600. AGE (Supplementary Fig. 9) revealed a single band with minor electrophoretic mobility shifts between QD600 wrapped with 5–40 nt A*, which may be due to a different set of conformations of the chimeric ssDNA when wrapped on the QD surface. Incubation of QDs with thiolated ssDNA

clearly generated products with a distribution of valences, consistent with previous observations[17]. We observed a shift of QD600 wrapped with 0 nt A* (without any ps-backbone) relative to the bare QD600, indicating some non-specific adsorption of 0 nt A*. To clarify the role of ps-backbone wrapping on the QD surface, ssDNA with only po-backbone (21 nt) and labelled with AF647 was incubated with either bare QD600 or QD600 wrapped with chimeric ssDNA composed of a 30 nt A* tract and a non-complementary po-backbone tract (QD600-30 nt A* with a non-complementary po domain). Steady-state emission spectroscopy showed significant emission quenching and QD-sensitized AF647 emission, indicating FRET for the AF647-bare QD mixture. In contrast, the emission intensity of QD600-30 nt A* with a non-complementary po domain was nearly unchanged (Supplementary Fig. 10), indicating the absence of FRET. AGE images in the

AF647 channel also showed a small amount of nonspecific DNA adsorption on bare QDs (Supplementary Fig. 10). Therefore, the A* tract can effectively prevent non-specific DNA adsorption, consistent with previous observations[55].

We then used all-atom molecular dynamics (MD) to build a molecular picture to interpret the impact of different lengths of A* tracts on DNA wrapping. All-atom MD simulations showed the various conformations of A* tracts wrapping on a 6 nm diameter ZnS nanoparticle surface (Supplementary Fig. 11). Once a single sulfur atom on the A* tracts anchored to the QD surface, the rest of the A* tract subsequently wrapped around the QD. The surface-bound chimeric ssDNA blocked the remaining active anchor sites, impeding additional ssDNA from approaching the surface of the QD[18]. All-atom MD simulations yielded a 51 nt A* radius of gyration ($R$g) of 6.2 nm. This $R$g corresponds to a Kuhn length of 2–3 nm and persistence length of ~5 nm (Supplementary Fig. 12) by self-avoiding walk polymer model, which is consistent with the low ionic strength environment simulated. By comparison, the persistence length of ssDNA in 14 mM MgCl$_2$ was measured to be approximately ~0.83–1.99 nm using atomic force microscopic imaging of two DNA origami rigid rods connected by a ssDNA chain[56]. MD simulations further showed that chimeric ssDNA conformations wrapped on the QD with different A* tracts lengths (5, 10, and 30 nt A*) and a fixed po-backbone length (23 nt) (Fig. 2f and Supplementary Movie 1). The distance between the 5′-end of the po-backbone and the QD surface increased when the A* tract length increased (49.2 ± 6.7 Å for 5 nt A*, 58.8 ± 10.6 Å for 10 nt A*, and 81.6 ± 10.0 Å for 30 nt A*), due to the increasing electrostatic repulsion between the ps and the po-backbones as more ps are adsorbed on the QD surface (Fig. 2f and Supplementary Fig. 12). The computational $R$g of the po domain also indicated that the 30 nt A* + 23 nt po ssDNA had a more extended chain conformation (Supplementary Fig. 12). When a shorter ps-backbone was on the QD surface, the po-backbone was more likely to be near to the QD surface, while the contact area between the 23 nt po-backbone and the QD significantly reduced when the A* tract length increased (Fig. 2g). The increased distance between the QD surface and po-backbone led to improved hybridization efficiency, approaching unity mainly when using chimeric ssDNA containing 30 nt A* for the DNA wrapping strategy (Fig. 2c, d). To further confirm the formation of monovalent QDs, QD dimers were constructed via hybridization and confirmed with transmission electron microscopy (TEM) imaging (Supplementary Fig. 13).

A different picture emerged for larger QDs. For QD660 (14 nm), a short ps tract length (5 nt A*) increased the probability of multiple copies of ssDNA wrapping on the QD surface, which led to a higher AF750/QD molar ratio and FRET efficiency compared with longer ps-backbone length (30 nt and 50 nt A*) (Fig. 2e). AGE of the ps-wrapped QDs could not distinguish multiple ssDNAs on QDs because the size of QD660 (14 nm) is significantly larger than QD600 (6 nm)[54], but we were still able to observe bands shift, indicating relatively higher electrophoretic mobility for shorter A* tracts (Supplementary Figs. 7 and 14) due to the multiple copies of DNA wrapping. Both steady-state and time-resolved emission spectroscopy were used to analyze QD660-AF750 FRET efficiency (Supplementary Figs. 4 and 6). QD donor emission intensities and lifetimes decreased concomitantly with decreasing A* tracts length from 50 to 5 nt, which may be attributed to multiple acceptors quenching the QD donor emission via FRET[51,57]. We also observed multiple wrapped chimeric ssDNAs in larger-sized QDs even as the A* tract length approached 50 nt, indicating that chimeric ssDNA with a 50 nt A* and a 23 nt po domain was insufficient to block the remaining active anchor sites on QDs with larger surface areas. We calculated the average donor-acceptor distance $R$ = 8.8 ± 0.8 nm for the QD660-AF750 FRET pairs, which was consistent with the mean donor-acceptor distance of $R$ = 7.8 ± 1.1 nm calculated using TEM (radius of QD660 (~7.1 ± 1.1 nm) + polyT spacer (~0.68 nm), Supplementary Fig. 2). MD simulations agreed with the

preceding experimental observations, with larger-sized NPs having more active anchor sites for DNA wrapping, increasing the probability of wrapping more than one chimeric ssDNA (Supplementary Fig. 15).

Compared with the monovalent QD-dye FRET, the FRET efficiencies of QD-thiol-dye FRET pairs calculated from steady-state measurements increased from 69 ± 3% to 81 ± 2%, from 59 ± 2% to 74 ± 7%, and from <5% to 40 ± 4% for QD600-thiol-AF647, QD630-thiol-AF647, and QD660-thiol-AF750, respectively, indicating the presence of multiple dye acceptors due to multivalent binding between QDs and 3′-thiolated-ssDNA (Supplementary Fig. 4). And the FRET efficiency of QD660-thiol-AF750 FRET pairs increased more significantly likely because the larger surface area could bind more ssDNAs. To further emphasize the advantage of the ssDNA wrapping strategy, commercially-available QD605 modified with streptavidin (streptavidin QD605) was used to fabricate the QD-streptavidin-biotin-AF647 FRET construct (Supplementary Figs. 16 and 17). We used the measured 72 ± 4% of FRET efficiency, the 7.0 nm Förster distance, and 9.8 nm donor-acceptor distance to estimate the number of AF647 dyes per streptavidin QD605 (Eq. (5), Methods). We calculated a value of 19 ± 4 AF647 dyes per streptavidin QD605, consistent with the result calculated from the three available biotin-binding sites on the surface of streptavidin QD605 and ~5–7 streptavidins bound to each streptavidin QD605, according to the manufacturer's specifications (Supplementary Fig. 17). This result highlighted several drawbacks of using the biotin-streptavidin conjugation strategy for precise ssDNA functionalization of QDs, including uncontrollable DNA binding and the larger donor-acceptor distances (~10 nm) introduced by the bulky streptavidin-biotin moieties.

## QD-DNA origami assemblies

Having investigated the interplay between A* tract length and QD size for valence control, we next sought to develop a general strategy for preparing valence-geocoded QDs. First, to test our ability to bind a single wireframe DNA origami within a QD, Pentagonal pyramid (Pep) wireframe origami objects with an overhang containing ps-backbone (30 nt A*) at the inner center or outer edge were used to fabricate Pep-30 nt A*-QD630 assemblies. For comparison with our valence-geocoding strategy, Pep with a biotin-modified overhang at the inner center or outer edge were designed to fabricate Pep-biotin-streptavidin-QD655 assemblies (Fig. 3a). QD630 and streptavidin-QD655 were used to fabricate the assemblies since they had similar sizes and were easy to observe in negative-stain TEM. AGE gel shift (Fig. 3b, c and Supplementary Figs. 18 and 19) and negative-stain TEM imaging (Fig. 3a, d–g and Supplementary Figs. 20–23) of the Pep-QD assemblies validated the assembly of the target DNA origami objects and different stoichiometric ratios of Pep per QD using chimeric ssDNA wrapping and streptavidin-biotin conjugation. By using chimeric ssDNA wrapping, each QD could only bind with a single Pep wireframe DNA origami object with an overhang at the inner center or outer edge. However, in the case of biotin-streptavidin conjugation, a streptavidin QD that contained multiple valences (active streptavidin) could bind one or two Pep with a biotin-modified overhang at the inner center, and up to three Pep with a biotin-modified overhang at the outer edge. Although the loading yields (Pep-QD/Pep) of Pep-30 nt A*-QD630 (inner center: 85%; outer edge: 87%) and Pep-biotin-streptavidin QD655 (inner center: 90%; outer edge: 86%) were similar, the yield of correct assemblies (monovalent Pep-QD/Pep) was nearly equal to the loading yield for the chimeric ssDNA wrapping strategy, compared with only 68% for biotin-streptavidin conjugation at the inner center, and 61% for biotin-streptavidin conjugation at the outer edge (Supplementary Fig. 24). We also noticed that a high proportion of divalent and trivalent Pep-QD assemblies could still form even when the Pep were incubated with fourfold excess streptavidin QD655 (Fig. 3c, f, g), which could be explained by the high-affinity

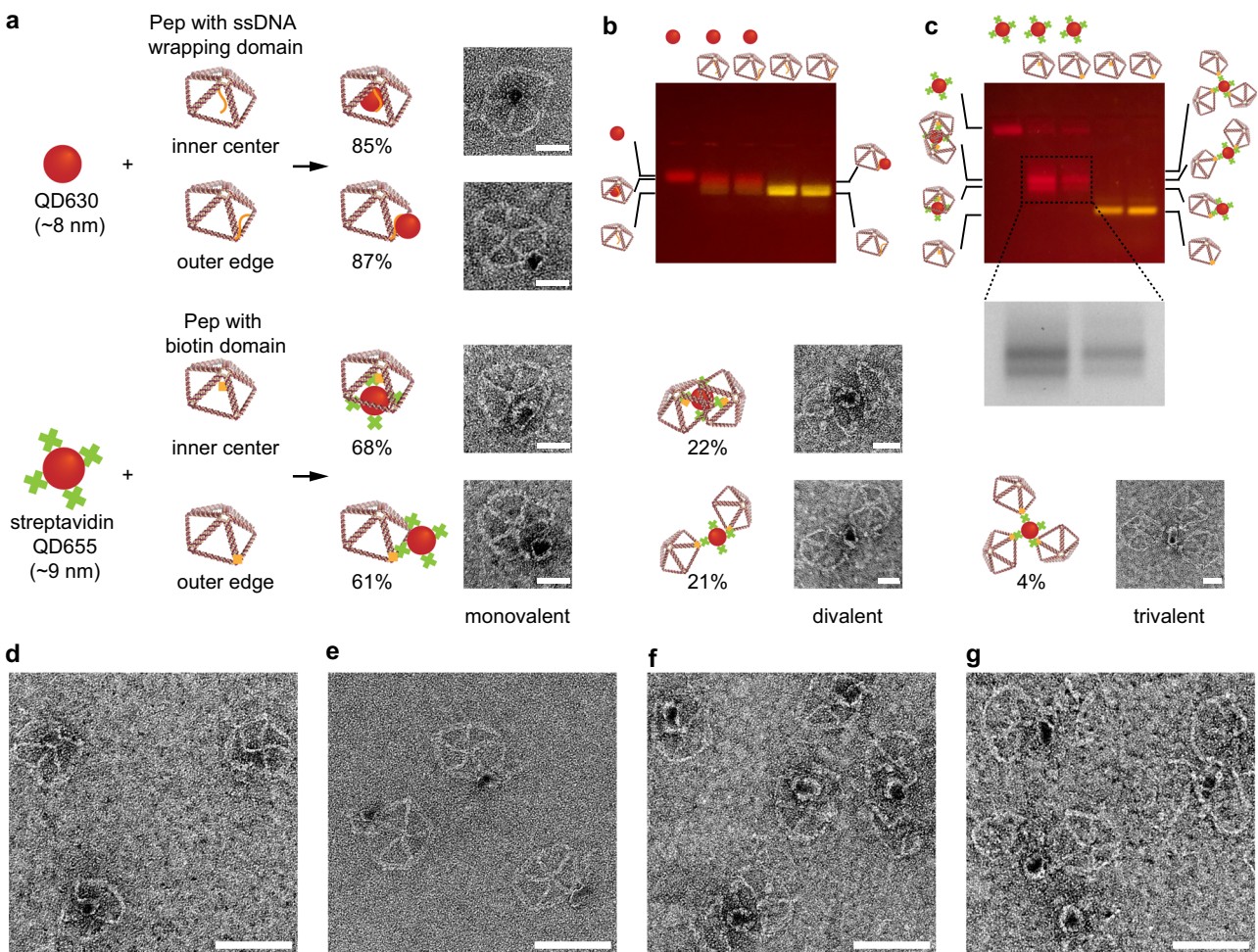

**Fig. 3 | Wireframe DNA origami-based chimeric ssDNA wrapping and biotin-streptavidin conjugation. a** Schematic and TEM images of pentagonal pyramid (Pep)-QD assemblies using chimeric ssDNA (30 nt A*) wrapping or biotin-streptavidin conjugation at the inner center and outer edge of Pep wireframe origami objects. The percentage of mono-, di-, and tri-valent Pep-QD assemblies were calculated from TEM images (308 and 337 assemblies for Pep with 30 nt A* domain at the inner center and outer edge, respectively; 257 and 265 assemblies for Pep with biotin domain at the inner center and outer edge, respectively). **b** AGE (0.8%) image (fourfold excess Pep) of QD630 alone, Pep-30 nt A*-QD630 (inner center), Pep-30 nt A*-QD630 (outer edge), Pep-30 nt A* (inner center), and Pep-30 nt A* (outer edge). **c** AGE (0.8%) image (fourfold excess QD) of streptavidin QD660 alone, Pep-biotin-streptavidin QD655 (inner center), Pep-biotin-streptavidin QD655 (outer edge), Pep-biotin (inner center), and Pep-biotin (outer edge). Representative TEM images of **d** Pep-30 nt A*-QD630 (inner center), **e** Pep-30 nt A*-QD630 (outer edge), **f** Pep-biotin-streptavidin QD655 (inner center), and **g** Pep-biotin-streptavidin QD655 (outer edge). (Scale bars: 20 nm **a**; 50 nm **d–g**.) Source data are provided as Source Data file.

biotin-streptavidin interaction ($K_d = 10^{-14}$ M)[58], because active streptavidin binding sites on QDs can still bind other Pep wireframe DNA objects after formation of Pep-QD assemblies provided they are accessible for binding. This result further demonstrated that monovalent ssDNA wrapping is crucial for accurate and efficient QD-DNA origami-based nanofabrication.

To prepare a library of valence-geocoded QDs of different sizes, a smaller tetrahedron (Tet) with 52 bp 6HB edges and a larger Pep with 63 bp 2HB edges were used to encapsulate QD600 and QD660, respectively, because these nanostructures have suitable internal cavity dimensions to fit single QDs (Fig. 4a, f and Supplementary Fig. 25). Each DNA origami nanostructure contained overhangs of controlled position, length, and sequence, which were used to control the respective valences of QDs. 30 nt A* and 50 nt A* were chosen as ssDNA binding domains for the QD600 and QD660 constructs, respectively, based on results of the previous section. AGE gel shift (Fig. 4d, i and Supplementary Figs. 26 and 27) and negative-stain TEM imaging (Fig. 4b, c, g, h, and Supplementary Figs. 28–31) of the Tet and Pep constructs validated the assembly of the target DNA origami objects and the incorporation of the single QD600 or QD660 in the

objects. The fidelities of the Tet-QD600 assemblies were analyzed using AGE. Due to the emission spectra overlaps of the nucleic acid fluorescence stain SYBR Safe and the QD600, these two species were distinguished using two fluorescence channels with different excitation wavelengths: the QD600 channel combined with SYBR Safe (blue light excitation) and the QD600-only channel (UV light excitation) (Fig. 4d and Supplementary Fig. 32). In contrast, because the QD660 and SYBR Safe channels have significantly different emission wavelengths, we were able to visualize the fidelity of the Pep-QD660 assembly with AGE and the QD660 and SYBR Safe channels individually (Fig. 4i and Supplementary Fig. 33). To further validate that a single QD was incorporated within the wireframe DNA origami object, we analyzed TEM images of Tet-QD600 and Pep-QD660 to validate interplanar spacing of the QD. The lattice fringes with interplanar spacing of 0.351 nm and 0.303 nm were measured using fast Fourier transform (FFT) and inverse FFT of selected areas of the TEM image (Fig. 4e, j and Supplementary Figs. 34 and 35), which were in agreement with the crystal lattice planes (111) and (200) corresponding to a bulk cubic zinc blende CdSe. The yield of the correct Tet-QD600 structure was up to ~91%. However, for QD660, the Pep with 50 nt A* ssDNA

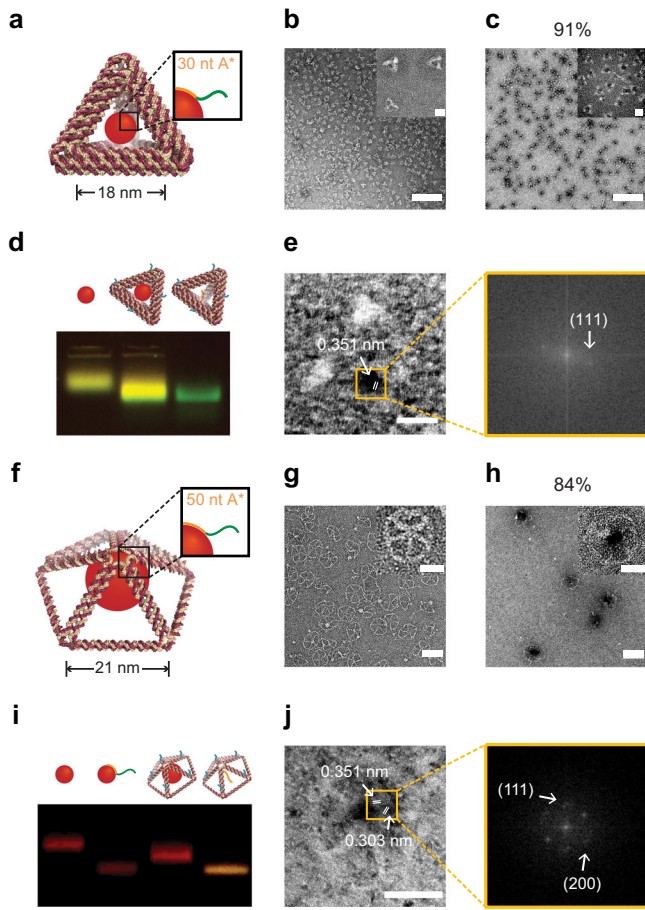

**Fig. 4 | Valence-geocoded QD using tetrahedron (Tet)-QD600 and pentagonal pyramid (Pep)-QD660 assemblies. a** Schematic of Tet-QD600 with 52 bp edges (~18 nm). Representative TEM images and HRTEM (inset) of **b** Tet wireframe DNA origami objects and **c** Tet-QD600. **d** AGE (0.8%) image of QD600 alone, Tet-QD600, and Tet wireframe DNA origami objects alone (yellow signal: QD600; green signal: QD600 and Tet wireframe DNA origami objects). **e** HRTEM and selected-area FFT pattern (orange box) of Tet-QD600. **f** Schematic of Pep-QD660 with 63 bp edges (~21 nm). Representative TEM images and HRTEM (inset) of **g** Pep wireframe DNA origami objects and **h** Pep-QD660. **i** AGE (0.8%) image of QD660 alone, QD660 wrapped with 50 nt A*, Pep-QD660, and Pep wireframe DNA origami objects alone (red signal: QD660; orange signal: Pep wireframe DNA origami objects). **j** HRTEM and selected-area FFT pattern (orange box) of Pep-QD660. The lattice fringes with d-spacing of 0.351 nm and 0.303 nm can be assigned to the (111) and (200) lattice planes of the cubic zinc blende CdSe. Scale bars: 200 nm for **b**–**c**; 50 nm for **g**–**h**; 20 nm for **b** inset, **c** inset, **g** inset, **h** inset and **j**; 10 nm for **e**. Source data are provided as Source Data file.

could lead to more than one wireframe DNA origami object wrapping (~20%) according to results of the previous section (Fig. 2e and Supplementary Fig. 36). To investigate this further, Pep was incubated with fourfold excess QD660. Compared with the 1:1 molar ratio reaction, incubation in a fourfold higher concentration of QD increased the loading efficiency substantially, with the yield of the correct structure increasing from ~59 to ~84% (Supplementary Fig. 36).

### Nanoscale spatial addressing of valence-geocoded QDs
To demonstrate spatial control of valence-geocoded QDs, we first fabricated a QD600-based FRET network with dyes. The Tet wireframe DNA origami object provides the ideal geometry to position dyes and QDs for efficient FRET. The overhangs of a Tet wireframe DNA origami object were designed to hybridize with 1–4 AF647 dyes on geocoded points and ~7 nm away from the center of the wireframe DNA origami

object (Fig. 5a). The assembly fidelities of the AF647 and wireframe DNA origami objects were analyzed using AGE with SYBR Safe and AF647 detection channels (Fig. 5b). In contrast, chimeric DNA complexes were designed to wrap a single QD and hybridize with 1–4 AF647 dyes, where the distances between QD and dyes were also ~7 nm (Fig. 5c and Supplementary Fig. 37). The formations of chimeric DNA complex and DNA complex-QD assemblies were confirmed using AGE imaging (Supplementary Figs. 38 and 39). Steady-state emission spectroscopy was used to analyze both the Tet-QD-dye and DNA complex-QD-dye-based FRET networks. QD600 emission intensities of both systems clearly decreased with increasing numbers of AF647 due to FRET, since the multiple acceptors provided multiple possible FRET pathways for the donor QD[51] (Supplementary Fig. 40). However, experimental results from the rigid Tet-QD-dye system were in agreement with theoretical calculations, whereas results from the flexible DNA complex-QD-dye system exhibited significant differences for 1–2 AF647 acceptors (*$P < 0.05$) (Fig. 5d). To further evaluate the importance of nanoscale spatial control of valence and the minimization of distance between QD and wireframe DNA origami objects, QD600-AF647-AF750 concentric multi-step FRET networks were fabricated using Pep wireframe DNA origami objects (Fig. 5e and Supplementary Fig. 41). For comparison, the same multi-step FRET networks were also fabricated using streptavidin QD605 instead of QD600 (Supplementary Figs. 42 and 43). These FRET networks are of interest because of their capability for single-vector multiplexed bioanalysis and imaging, which offers several advantages, such as more efficient use of the spectrum and one-shot excitation[9,59]. However, earlier concentric FRET networks relied on the 1:*M*:*N* (*M* and *N* refer to the statistical mean number of fluorophores) stoichiometry between a QD donor, a dye relay, and an acceptor[60–64] or the *M*:1:*N* stoichiometry between a lanthanide donor, a QD relay, and a dye acceptor[51,65], because these previous methods failed to achieve 3D spatial addressing and valence control of QDs. Thus, the fluorophores (lanthanide complexes and dyes) were randomly distributed on the QD surface, and the numbers of conjugated fluorophores were determined from statistical mean values. Here we showed three types of concentric multi-step FRET networks can be achieved with fixed stoichiometry (1:1:1) between fluorophores due to the 3D spatial addressing and valence control of QDs (Fig. 5e). Unlike the Tet wireframe DNA origami object, the Pep wireframe DNA origami object allowed us to fabricate asymmetric QD-dye assemblies. The AF647 was incorporated into the Pep during the DNA wireframe origami folding, and the assembly fidelities were analyzed using AGE with SYBR Safe and AF647 detection channels (Supplementary Fig. 43). The overhangs of a Pep wireframe DNA origami object were designed to hybridize with AF750 dyes at close (~7.1 nm), medium (~7.8 nm) or far (~12.2 nm) distances to AF647 but similar distances to the QD (~9–10 nm) (Fig. 5e and Supplementary Figs. 41 and 44). Experimental results from the Pep-30 nt A*-QD600-AF647-AF750 multi-step FRET networks were in agreement with theoretical calculations (Fig. 5f, g, and Supplementary Fig. 45), whereas results from Pep-biotin-streptavidin QD605-AF647-AF750 FRET networks exhibited considerably lower energy-transfer efficiency between initial donor and relay, most likely due to the bulky streptavidin tags, which fail to realize the multi-step FRET process (Supplementary Figs. 42 and 45). These results further demonstrated that our approach could maintain distances between additional functionalities and the QD surface at 2.8 nm for Pep-30 nt A*-QD600 assemblies, whereas the inter-dye-QD surface distance was up to ~7.4 nm for the Pep-biotin-streptavidin-QD605 (Supplementary Fig. 42).

The preceding demonstration of valence-geocoded QDs also provides a synthetic methodology for the programmed self-assembly of QD-based colloidal molecules on wireframe DNA origami objects. Unlike the Tet wireframe DNA origami object, the Pep wireframe DNA origami object has a larger size that allows for the immobilization of additional QDs and provides additional sites to fabricate

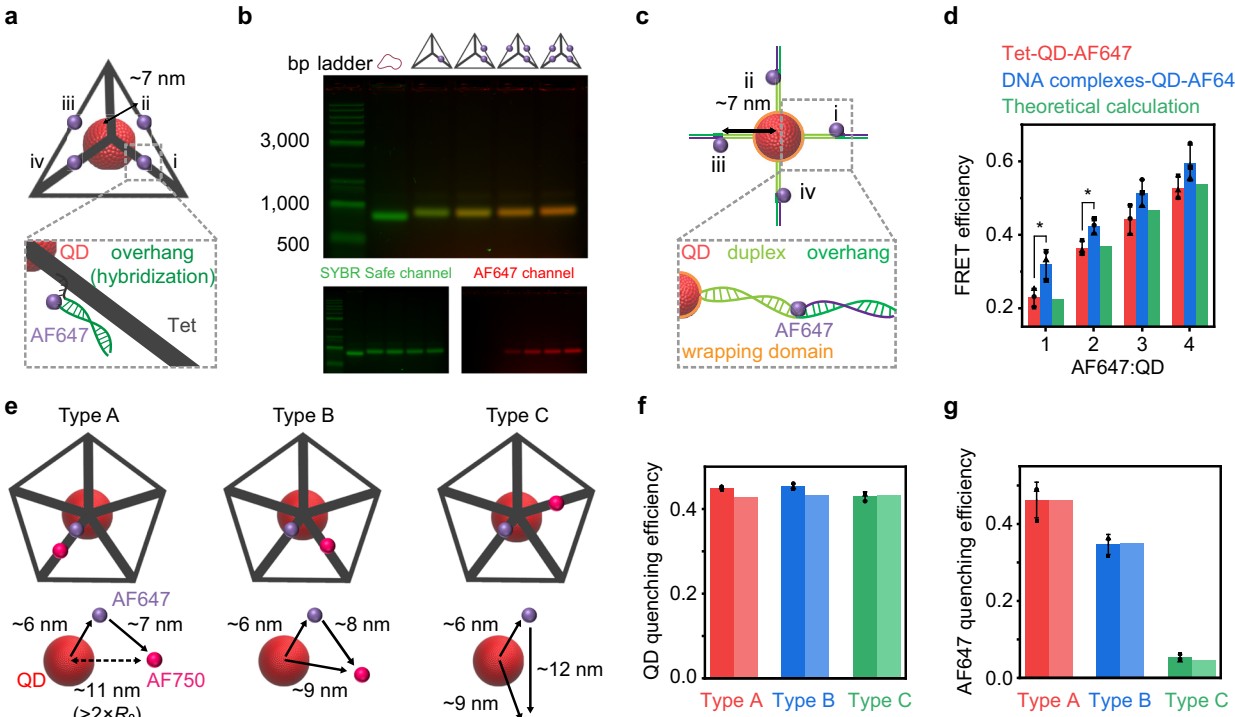

**Fig. 5 | Valence-geocoded QD-based energy transfer circuits. a** Schematic of the Tet-QD-based FRET network and the overhang design (the distance for each de-excitation pathway is ~7 nm). **b** AGE (1.5%) image for 1–4 AF647-labeled Tet wire-frame DNA origami objects. (Green: SYBR safe channel, Red: AF647 channel). **c** Schematic of the DNA complex-QD-based FRET network. **d** FRET efficiencies as a function of acceptors calculated from the Tet-QD-dye system, the DNA complex-QD-dye system, and Förster theory (Eq. (5)). *P* values are from Student's *t* test

(*$P < 0.05$); all *P* values are provided in Supplementary Table 3. **e** Schematics of three types of Pep-QD-based multi-step FRET networks and predictions of distances between initial donor (QD), relay (AF647), and acceptor (AF750). **f** QD quenching efficiencies and **g** AF647 quenching efficiencies calculated from the Pep-QD based multi-step FRET networks (darker bar), and Förster theory (lighter bar) (Eqs. (6)–(13)). Error bars in **d**, **f**, **g** represent standard deviations of the mean (*n* = 3 replicates per group). Source data are provided as Source Data file.

asymmetric QD superstructures. To test our ability to control spatial relationships of QDs, Pep-based QD trimers with specific geometries containing distinct angles θ, denoted Type A (θ = 60°) and Type B (θ = 120°), were fabricated. The overhangs of the Pep wireframe DNA origami object were designed with a ps sequence to rigidly attach QDs to the wireframe DNA origami object and maintain a close distance between the QD and wireframe DNA origami object (Fig. 6a). TEM showed that valence-geocoded QD-based Type A and Type B were fabricated with high fidelity (75%) (Fig. 6c, d). The distributions of bond angles of valence-geocoded QD-based trimer Type A and Type B were calculated using more than 200 randomly selected QD trimers in TEM images. Angles were largely distributed between 50°–100° for Type A (87%) and 100°–160° for Type B trimer QDs (87%) (Fig. 6b). Similarly, for comparison, DNA complex-based QD trimers with distinct angles θ, denoted Type C (θ = 90°) and Type D (θ = 180°), were fabricated (Fig. 6h). However, the bond angles of both trimer Type C and Type D exhibited broad variation, which could not be distinguished effectively (Fig. 6i). Moreover, TEM showed that trimer Type C and Type D gave low fabrication yields of the target structure due to the use of equimolar stoichiometry and lack of effective purification methods (Fig. 6j, k), and HRTEM showed that spatial control of the DNA complex-based trimer Type D failed due to the "floppy" duplex structure (Fig. 6l).

To demonstrate the generality of our valence-geocoded approach, we fabricated QD trimers, tetramers, pentamers, and hexamers on wireframe DNA origami objects using ssDNA wrapping. TEM images of the fabricated QD-based colloidal molecules showed high fidelity of self-assembly of the target structures (Fig. 6c–g), with calculated yields of 75% for the trimer, 68% for the tetramer, 56% for the pentamer, and 48% for the hexamer (Supplementary Figs. 46–51). The

preceding examples demonstrate that the ssDNA wrapping approach combined with DNA origami self-assembly of wireframe structures offers not only precise programming of valences, but also of spatial positions determining relative angular orientations of QD centers. In the current implementation, QD-based colloidal molecules were fabricated using a single type of QD because we used wireframe DNA origami objects with ps-backbone overhangs to wrap the QDs directly. Future work could extend our approach to include multiple QDs with different band gaps that are geocoded using different staple overhangs on the wireframe DNA origami.

## Discussion

We demonstrated a general strategy to fabricate valence-geocoded QDs using ssDNA wrapping and self-assembly of wireframe DNA origami objects. ssDNA wrapping behaviors were examined for various QD sizes and ps-backbone lengths using a hybrid experimental-computational approach. We found experimentally that the length of the ps segment of chimeric ssDNA and QD size were important parameters to consider for efficient hybridization of the po segment, and for control of the number of chimeric strands per QD, theoretically due to the conformations of ssDNA wrapped on a QD, as supported by MD simulations. We showed that this chimeric ssDNA wrapping strategy could be applied to wireframe DNA origami objects of different geometries using ps-backbone-modified overhangs. Our approach minimizes the distance between the QD and wireframe DNA origami object, as supported by FRET measurements, successfully eliminating bulky biotin-streptavidin tags that are often used to attach QDs to DNA nanostructures[15,41]. The combination of these advances offers a generalized approach to fabricating QDs immobilized on wireframe DNA origami scaffolds in order to control QD valences and spatial positions

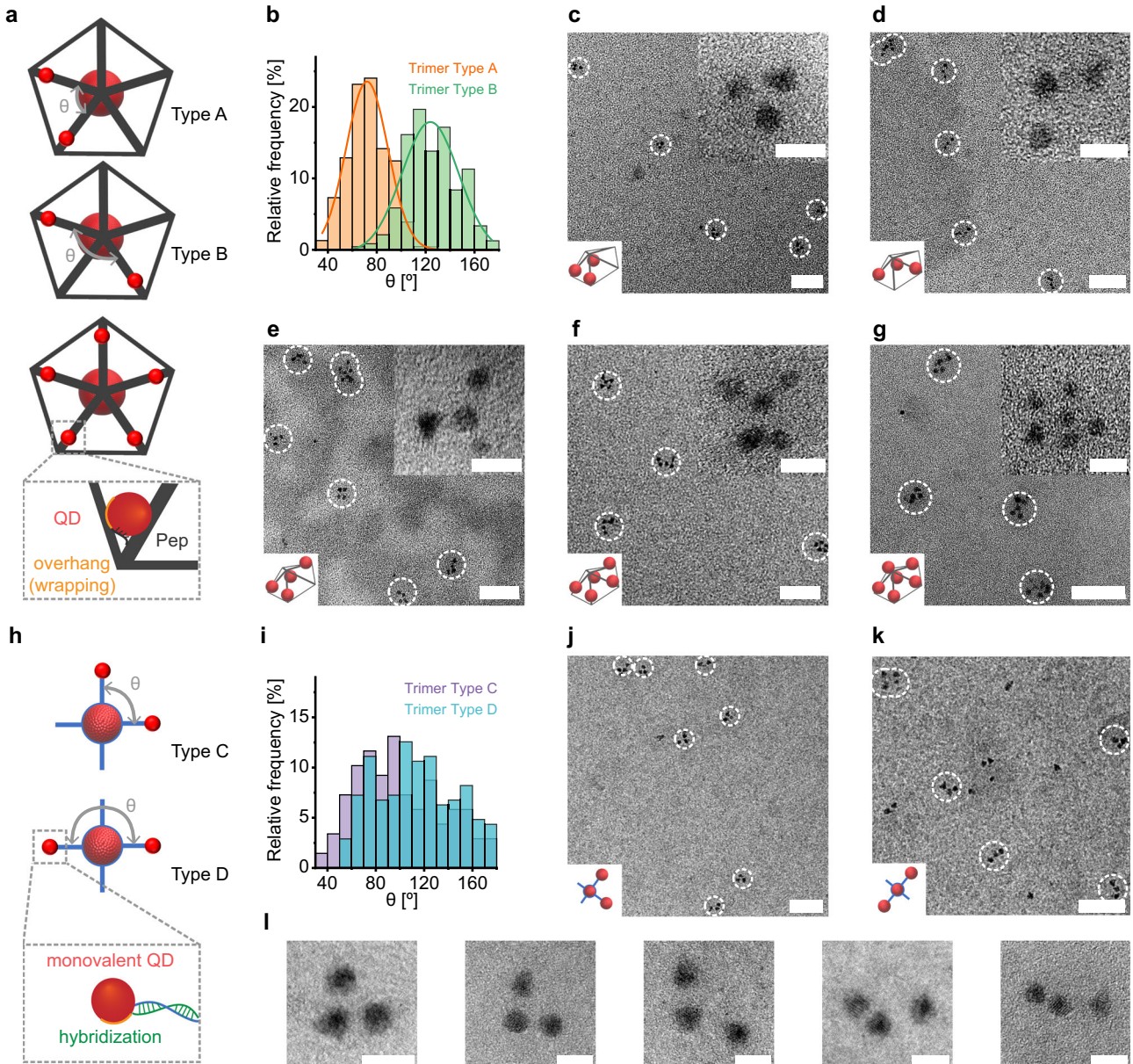

**Fig. 6 | Valence-geocoded QD-based colloidal molecules. a** Schematic of Pep-based QD assemblies, the theoretical angle of θ for QD trimer Type A and Type B, and the overhang design. **b** Distributions of bond angles of randomly selected QD trimer in TEM images (233 NPs for Type A, 238 NPs for Type B, solid curve are Gaussian fitting of the distribution). TEM, HRTEM (inset upper right) and schematic (inset bottom left) image for QD trimer **c** Type A and **d** Type B, **e** QD tetramer, **f** QD pentamer and **g** QD hexamer. **h** Scheme for DNA complexes-based QD assemblies, the theoretical angle of θ for QD trimer Type C and Type D. **i** Distributions of bond angles of randomly selected QD trimer in TEM images (206 NPs for Type C, 207 NPs for Type D). TEM and schematic (inset bottom left) image for QD trimer **j** Type C and **k** Type D. **l** HRTEM image for QD trimer Type D. (white circles indicate correctly formed assemblies. Scale bars: 50 nm for **c–g**, **j**, **k**; 10 nm for inset **c–g**, **i**). Source data are provided as Source Data file.

that determine relative orientations of QD centroids in an arbitrary manner.

The use of ssDNA wrapping combined with wireframe DNA origami objects to engineer QD valences offers three advantages over the state-of-the-art: (1) ssDNA wrapping limits the formation of multivalent QDs that could otherwise promote the binding of multiple DNA origami objects to a single QD; (2) 3D wireframe DNA origami provides spatial position control of valence that offers better connectivity and enables in principle the self-assembly of higher-order structures with multiple QD-wireframe DNA origami objects[41]; and (3) QDs can also in principle be assembled on 2D or 3D wireframe DNA origami objects that may also contain multiple copies of dyes, peptides, nucleic acids, drugs, or proteins, organized in two or three-dimensional space.

Taken together, the use of wireframe DNA origami objects and nanoparticle surface modifications should help translate nanoscale DNA origami design strategies into functional hybrid nanoparticle materials for programmable nanoscale excitonic and photonic materials[66], where discrete control over the individual components is critical for achieving desired device performance.

## Methods
### General materials
CdSe/ZnS core/shell quantum dots (catalog number: 900218(QD600), 900220(QD630) and 900249(QD660)), 3-MPA (≥99%, catalog number: M5801), zinc acetate (99.99% trace metals basis, catalog number: 383317), O-(2-Mercaptoethyl)-O′-methyl-hexa(ethylene glycol) (mPEG)

(≥95% (oligomer purity), catalog number: 672572), trioctylphosphine oxide (TOPO) (ReagentPlus®, 99%, catalog number: 223301), tetra-butylammonium bromide (TBAB) (ACS reagent, ≥98.0%, catalog number: 426288) were purchased from Sigma-Aldrich. Streptavidin Qdot® conjugates (CdSe/ZnS core/shell QD, catalog number: Q10151MP) were purchased from Thermo Fisher Scientific. Basic Agarose (catalog number: IB70070) was purchased from IBI Scientific. TAE 10× buffer (pH 8.3 ± 0.1 RNase-/DNase- and protease-free, catalog number: 46-010-CM) and PBS 1× buffer (pH 7.4 ± 0.1, without calcium and magnesium, catalog number: 21-040-CM) were purchased from Corning®. All DNA oligonucleotides were purchased from Integrated DNA Technologies (IDT; Coralville, IA) with standard desalting (for DNA oligonucleotides with lengths <70 nt), PAGE (for DNA oligonucleotides with lengths exceeding 70 nt), or HPLC (for dye-modified DNA oligonucleotides) as purification method. All DNA oligonucleotides were received as dry pellets. Sodium chloride (5 M, catalog number: AM9760G), MgCl₂ (1 M, catalog number: AM9530G) and Tris (1 M, pH 8.0, catalog number: AM9855G) were purchased from Life Technologies Corporation DBA Invitrogen. The custom circular DNA scaffold phPB84 and pF1A were prepared according to previous work[67].

### Aqueous QD with tunable surface charge

Aqueous QD was prepared as described previously, with minor modifications[17,18]. Briefly, 80 μL of QD (5 mg/mL, QD600, QD630 and QD660 in chloroform) were incubated with 160 μL of TOPO (1 g/10 mL in chloroform) and 160 μL of chloroform at 25 °C and shaken at 1200 rpm using a thermal mixer (Thermo Fisher; catalog number: 13687711). After 30 min, 20 μL of TBAB (0.3 M in chloroform) was added to this mixture. After an additional 30 min of incubation and shaking, 400 μL Zn-MPA in NaOH (11 mM in 0.2 M aqueous NaOH) was added. The mixture was briefly vortexed and centrifuged with a benchtop mini-centrifuge for 30 s, and the aqueous layer was recovered. The vortexing and centrifugation steps were repeated until all aqueous layers were collected. MPA-QDs were purified to remove excess Zn-MPA and concentrated using an ultracentrifugal filter (Amicon 30 kDa) five times at 6000 × g for 10 min for each centrifugation step. After purification, the MPA-QDs were diluted to 500 μL with nuclease-free water and incubated with 5, 10, and 20 μL of mPEG for 4 days at room temperature. The MPA/mPEG-QDs were purified and concentrated using an ultracentrifugal filter (Amicon 30 kDa) five times (6000 × g, 10 min), and then buffer exchanged into 10 mM Tris using a NAP-5 desalting column (GE Healthcare). QD concentrations were determined using measured absorbances at 350 nm (extinction coefficients of QDs that emit at 600 nm, 630 nm and 660 nm are 3,000,000 M⁻¹ cm⁻¹, 10,600,000 M⁻¹ cm⁻¹ and 29,000,000 M⁻¹ cm⁻¹, respectively).

### DNA wrapped QD and QD-dye FRET pairs

QDs with neutral surface charge were incubated with ps-backbone-modified ssDNA or 3′-thiolated ssDNA overnight for DNA wrapping or conjugation (molar ratio was 1:10). Briefly, ten 2 μL portions (20 μL total) of chimeric ssDNA (0.1 mM) with different ps tract length (5, 10, and 30 nt A* for QD600; 5 nt A*, 10 nt A*, and 30 nt A* for QD630; 5 nt A*, 30 nt A*, and 50 nt A* for QD660) and fixed po tract length (23 nt) or 3′-thiolated ssDNA were added to 500 μL of 400 nM QD with neutral surface charge in 10 mM Tris. After overnight incubation, the mixture was purified and concentrated using ultracentrifugal filter (Amicon 50 kDa) five times at 6000 × g for 10 min for each purification cycle. Finally, purified QD-DNA concentrations were determined by measuring the absorbance of samples at 350 nm. To fabricate QD600-AF647 and QD630-AF647 FRET pairs, QD600-5 nt A*, QD600-10 nt A*, QD600-30 nt A*, QD600-thiol, QD630-5 nt A*, QD630-10 nt A*, QD630-30 nt A*, and QD630-thiol were incubated with complementary DNA labeled with AF647. To fabricate QD660-AF750 FRET pairs, QD660-5 nt A*, QD660-30 nt A*, QD660-50 nt A*, and QD660-thiol were incubated with complementary DNA labeled with AF750. After 2 h incubation, the

fluorescence emission spectra of QD alone and in the presence of AF647 or AF750 were recorded. The QD-dye mixtures were purified to remove excess AF647- or AF750-labeled ssDNA using ultracentrifugal filter (Amicon 100 kDa) five times at 6000 × g for 10 min for each purification cycle. Then the QD/AF647 and QD/AF750 molar ratios were determined by absorption spectra. All DNA sequences are summarized in Supplementary Table 4.

### Effect of phosphorothioate (ps) tract length

To 100 μL of neutral-surface-charge QD600 (100 nM in 10 mM Tris), 5 μL of the ssDNA (0.1 mM) with various ps tract length (0, 5, 10, 20, 30, and 40 nt A*) and fixed total tract length (51 nt) or 5′-thiolated-ssDNA (51 nt) was added (divided into ten additions). After overnight incubation, 15 μL of mixture was combined with 3 μL of 6× loading buffer (NEB) and loaded to a 0.8% agarose gel with 1× TAE. Each gel was run at 65 V in 1× TAE buffer at 4 °C for 40 min or 80 min. Gels were then visualized under blue light.

To 100 μL of neutral-surface-charge QD660 (10 nM in 10 mM Tris), 5 μL of the ssDNA (10 μM) with various ps tract length (5, 10, 20, 30, and 40 nt A*) and fixed tract length (51 nt) was added (divided into ten additions). After overnight incubation, the mixture was applied to an E-Gel Electrophoresis System (Thermo Fisher Scientific) using E-Gel™ NGS™ 0.8% Agarose Gels with SYBR Safe (Invitrogen™, catalog number: A25798). All DNA sequences are summarized in Supplementary Table 5.

### DNA origami folding

Tetrahedron (Tet) wireframe DNA origami objects with six-helix bundle edges were folded in a solution of 50 nM scaffold (phPB84), 500 nM staples, 1× TAE, and 12 mM MgCl₂ and annealed over the course of 12 h (95 °C for 5 min, 80 °C down to 75 °C for 5 min/°C, 75 °C down to 30 °C for 15 min/°C, 30 °C down to 25 °C for 10 min/°C) on a Bio-Rad T100 thermocycler (Hercules, CA). Folding was initially checked by agarose gel mobility shift assays. Pentagonal pyramid (Pep) wireframe DNA origami objects with two-helix bundle edges were folded in a solution of 30 nM scaffold (pF1A), 300 nM staples, 1× TAE, and 12 mM MgCl₂ and annealed over the course of 2 h (95 °C for 5 min, 80 °C down to 75 °C for 0.8 min/°C, 75 °C down to 30 °C for 2.4 min/°C, 30 °C down to 25 °C for 1.6 min /°C) on a Bio-Rad T100 thermocycler (Hercules, CA). Folded sample (15 μL) was combined with 3 μL of 6× loading buffer (NEB) and loaded to a 1.5 % agarose gel with 1× TAE and 12 mM MgCl₂ and 1× SYBR Safe (ThermoFisher, Waltham, MA). Each gel was run at 65 V for 180 min in 1× TAE with 12 mM MgCl₂ at 4 °C. Gels were then visualized using Typhoon™ FLA 7000 biomolecular imager. Folded wireframe DNA origami objects were purified from staples and folding buffer by using an ultracentrifugal filter (Amicon 100 kDa). Tetrahedron and pentagonal pyramid origami wireframe were exchanged into buffer composed of 1× TAE with 12 mM MgCl₂ by centrifugation at 2000 × g for 25 min at 25 °C, respectively, diluted approximately tenfold, and reconcentrated for a total of five times using the ultracentifugal filter. Finally, wireframe DNA origami objects concentration was determined by absorbance at 260 nm. All DNA sequences are summarized in Supplementary Tables 6–8.

### DNA origami biotin-streptavidin QD and DNA origami 30 nt A*-QD assemblies

To assemble Pep-biotin-streptavidin-QD655, pentagonal pyramid wireframe DNA origami objects with a biotin domain at the inner center or outer edge were incubated with Streptavidin QD655 (molar ratio was 1:4) at room temperature for 2 h. The mixture (15 μL) was then combined with 3 μL of 6× loading buffer (NEB) and loaded to a 0.8% agarose gel with 1× TAE and 12 mM MgCl₂ and 1× SYBR Safe (Thermo Fisher Scientific, Waltham, MA). Each gel was run at 65 V for 2 h in 1× TAE with 12 mM MgCl₂ at 4 °C. Gels were then visualized under blue

light. To assemble Pep-30 nt A*-QD630, pentagonal pyramid wireframe DNA origami objects with ps-backbone-based ssDNA wrapping domain at the inner center or outer edge were incubated with QD630 (molar ratio was 1:4 or 4:1) at room temperature overnight. The mixture (15 μL) was then combined with 3 μL of 6× loading buffer (NEB) and loaded to a 0.8% agarose gel with 1× TAE and 12 mM MgCl₂ and 1× SYBR Safe (Thermo Fisher Scientific, Waltham, MA). Each gel was run at 65 V for 1 h in 1× TAE with 12 mM MgCl₂ at 4 °C. Gels were then visualized under blue light. All DNA sequences are summarized in Supplementary Tables 6 and 8.

### Tet-QD600 and Pep-QD660 assemblies

For Tet-QD600, Tet wireframe DNA origami objects with QD wrapping domain were incubated with QD600 (molar ratio was 1:4, in 1× TAE with 20 mM MgCl₂) and undergo a secondary annealing protocol (55 °C for 10 min, 55 °C down to 45 °C for 2 min/°C, 45 °C down to 25 °C for 10 min/°C). To assemble Pep-QD660, pentagonal pyramid wireframe DNA origami objects with a QD wrapping domain were incubated with QD660 (molar ratio was 1:4) at room temperature overnight. The mixture (15 μL) was then combined with 3 μL of 6× loading buffer (NEB) and loaded to a 0.8 % agarose gel with 1× TAE and 12 mM MgCl₂ and 1× SYBR Safe (ThermoFisher, Waltham, MA). Each gel was run at 65 V for 40 min in 1× TAE with 12 mM MgCl₂ at 4 °C. Gels were then visualized under blue light and UV light for Tet-QD600, and under blue light for Pep-QD660.

### Valence-geocoded QD based energy-transfer circuits (QD-dye assemblies)

Tet-QD600-based FRET network: Tet wireframe DNA origami objects with 1–4 overhangs were incubated with complementary ssDNA labelled AF647 (molar ratio was 1:10, in 1× TAE with 20 mM MgCl₂) at room temperature. After overnight reaction, the mixture (15 μL) without purification was combined with 3 μL of 6× loading buffer (NEB) and loaded to a 1.5% agarose gel with 1× TAE and 12 mM MgCl₂ and 1× SYBR Safe (ThermoFisher, Waltham, MA). Gel was run at 65 V for 3 h in 1× TAE with 12 mM MgCl₂ at 4 °C. Gels were then visualized under a SYBR Safe channel (excitation: 473 nm, emission: LP filter 520 nm) and an AF647 channel (excitation: 635 nm, emission: LP filter 670 nm) using Typhoon™ FLA 7000. For the Tet-QD600-based FRET network, the mixture was purified to remove excess AF647 using an ultracentrifugal filter (Amicon 100 kDa) five times (2000 × g, 25 min). Then the Tet DNA NPs with 1–4 AF647 dyes attached were incubated with QD600 (molar ratio was 2:1) using the previous protocol. After overnight reaction, the fluorescence emission spectra were recorded. All DNA sequences are summarized in Supplementary Table 7.

Pep-QD-AF647-AF750-based multi-step FRET networks: Pep wireframe DNA origami objects with spatially addressable overhangs were incubated with complementary ssDNA labelled AF750 (molar ratio was 1:2, in 1× TAE with 12 mM MgCl₂) at room temperature. After overnight reaction, the Pep-AF647-AF750 were then incubated with QD600 (molar ratio was 2:1) at room temperature. After 2 h reactions, fluorescence emission spectra were recorded. All DNA sequences are summarized in Supplementary Table 8.

### Valence-geocoded QD based colloidal molecules (QD-QD assemblies)

Pep wireframe DNA origami objects with ps-backbone-modified overhangs on geocoded points (trimer Type A, trimer Type B, tetramer, pentamer, and hexamer) were incubated with excess QD600 (molar ratio was 1:5). After overnight reaction, the mixtures were purified using toehold-mediated strand displacement based magnetic separation as described previously[68,69]. Briefly, 1 μL of 3′biotin-labeled strand sequence (10 μM) was added with a molar ratio of 1:2 to 50 μL of colloidal molecules (100 nM) with specific overhangs (detail in Supplementary Information). After overnight reaction, the mixture was

incubated with 5 μL of Dynabeads™ M-270 Streptavidin (10 mg/mL, ≥950 pmoles free biotin/mg beads) (Invitrogen™, catalog number: 65306). After overnight reaction and shaking, the colloidal molecules and magnetic bead assemblies were captured by DynaMag™-PCR Magnet (Invitrogen™, catalog number: 492025) and washed using 1× TAE and 12 mM MgCl₂. We repeated this process until no fluorescence could be observed in the supernatant. Then, 50 μL of bead invader sequence (2 μM in 1× TAE and 12 mM MgCl₂) was added (molar ratio of colloidal molecules/bead strand/bead invader was 1:2:20). After overnight reaction and shaking, the magnetic beads were captured by DynaMag™-PCR Magnet, and the supernatants were collected. All DNA sequences are summarized in Supplementary Table 9.

### DNA complex-QD-based FRET network and trimers

Equimolar concentrations of the four types of chimeric DNA were mixed together, and annealed using the following protocol: 95 °C for 5 min, 90 °C for 10 min, then directly cooled down to 4 °C in refrigerator. To prepare QD-DNA complex constructs with valence (I)–(IV), the prepared DNA complex with valence (I)–(IV) were incubated with QD (molar ratio was 1.2:1 in 1× PBS) overnight. To fabricate DNA complex-QD-dye FRET pairs, DNA complex-QD with valence (I)–(IV) were incubated with complementary DNA labeled with AF647 in 1× PBS buffer. After 2 h incubation, the fluorescence emission spectra of complex-QD alone and in the presence of AF647 were recorded. To fabricate the DNA complex-based QD trimers, DNA complex-QD with valence (II) (Type C and Type D) were incubated with complementary DNA labeled monovalent QD in 1× PBS buffer overnight. All sequences are summarized in Supplementary Table 10.

### Electron microscopy and spectroscopic characterization

Structural characterization of the QDs, wireframe DNA origami objects, and QD-wireframe DNA origami objects assemblies were carried out using a ThermoFisher FEI Tecnai Spirit Transmission Electron Microscopy operating at 120 kV. For QDs with organic ligands, 10 μL of QDs (50 μg/mL) were drop casted on 400 mesh carbon film square grids (Fisher Scientific, catalog number: 5024891). For aqueous compatible QDs, 10 μL of QDs (50 nM) were drop casted on 400 mesh carbon film square grids (Fisher Scientific, catalog number: 5024891). For wireframe DNA origami objects and QD-wireframe DNA origami objects assemblies, 10 μL of wireframe DNA origami objects with or without attached QDs (20 nM) were adsorbed on glow-discharged 400 mesh carbon film square grids and stained by 2% aqueous uranyl formate solution containing 25 mM of NaOH. For QD colloidal molecules, 10 μL of samples (1 nM) were adsorbed on glow-discharged 400 mesh carbon film square grids.

Absorbance spectra were measured using an Evolution 260 Bio UV-Vis spectrophotometer (Thermo-Fisher) and steady-state emission spectra ($\lambda_{ex}$ = 450 nm) were measured using a FluoroMax-4C (Horiba Jobin Yvon). Extinction coefficient of QDs were estimated by the first extinction absorption peak, and calculated according the empirical formula from the literature[70]. Quantum yields of QDs were determined using the relative quantum yield determination method with rhodamine 101 in spectroscopic-grade ethanol as standard ($\lambda_{ex}$ = 480 nm, $\Phi_s$ = 0.92)[71].

All fluorescence spectra are corrected for lamp fluctuations and detector sensitivity (or S1c/R1c detector setting). The absorbance and fluorescence spectra of all the samples were measured in 10 mm path length quartz micro cuvettes (Millipore Sigma, catalog number: Z802662). For the ensemble fluorescence lifetime measurements, the excitation was generated by a tunable fiber laser (FemtoFiber pro, Toptica Photonics, 2.5 MHz repetition rate, 490 nm, 4 nm full-width half maximum (fwhm)), passed through a pinhole, and directed into a home-built confocal microscope. The excitation was focused by an oil-immersion objective (UPLSAPO100XO, Olympus, NA 1.4) onto the samples immobilized on a coverslip (Electron Microscopy Sciences,

catalog number: 72290-12). The coverslip was mounted on a piezo-stage controlled by a home-written Labview-based software. The emission of the sample was collected through the same objective and separated from the excitation by using band pass filter (FF01-600/52-25, Semrock) and (FF02-685/40-25, Semrock). The average power of excitation was ~1 μW. Emission was detected by a silicon-based single photon counting avalanche photodiode (SPCM-AQRH, Excelitas Technologies). A time-correlated single-photon counting (TCSPC) module (Time Tagger 20, Swabian Instruments) was used to collect photons.

## FRET calculations

The overlap integral ($J$) and Förster distance ($R_0$) were calculated using Eq. (1) and Eq. (2)[72].

$$J = \int \bar{I}_D(\lambda) \varepsilon_A(\lambda) \lambda^4 d\lambda \tag{1}$$

where $\bar{I}_D(\lambda)$ is the area-normalized emission spectrum of the donor, $\varepsilon_A(\lambda)$ is the molar absorptivity spectrum of the acceptor in $M^{-1}cm^{-1}$, and $\lambda$ is the wavelength in nm[72].

$$R_0 = 0.0211 \left[ \kappa^2 \Phi_D n^{-4} J(\lambda) \right]^{\frac{1}{6}} \text{ (in nm)} \tag{2}$$

where $\kappa^2$ is the orientation factor ($\kappa^2 = 2/3$ due to dynamic averaging within all FRET pairs as justified by the flexible attachment of AF647/AF750 to the DNA and the isotropic emission of QD), $\Phi_D$ is the quantum yield of the donor, and $n = 1.35$ is the refractive index of the surrounding medium. The molar extinction coefficients for AF647 and AF750 were provided by the suppliers.

FRET efficiencies were calculated using emission intensities (Eq. (3)) and decay times (Eq. (4))[72],

$$E_{FRET} = 1 - \frac{I_{DA}}{I_D} \tag{3}$$

where $I_{DA}$ is the emission intensity of the QD-dye FRET pairs and $I_D$ is the emission intensity of QD alone[72].

$$E_{FRET} = 1 - \frac{\tau_{DA}}{\tau_D} \tag{4}$$

where $\tau_{DA}$ is the average emission lifetime of the QD-dye FRET pairs and $\tau_D$ is the average emission lifetime of QD alone.

In the case of FRET from one QD to $n$ equidistant AF647 dyes, the FRET efficiency can be calculated using Eq. (5)[57],

$$E_{FRET} = \frac{nR_0^6}{nR_0^6 + R^6} \tag{5}$$

In the case of QD-AF647-AF750 FRET networks, quantifying competing (QD-AF647 and QD-AF750) and sequential (AF647-AF750) energy transfer pathways requires the use of rates instead. The FRET rate, $k_{D \to A}$ and the corresponding FRET efficiency can be calculated using Eq. (6) and Eq. (7)[9],

$$k_{D \to A} = k_D \left( \frac{R_0}{R} \right)^6 \tag{6}$$

$$E_{D \to A} = \frac{k_{D \to A}}{k_D + k_{D \to A}} \tag{7}$$

where $k_D$ is the natural excited-state relaxation rate of the donor. The relative rate, $\gamma_{D \to A}$, can be calculated using Eq. (8). Using the relative

rates calculated in Eq. (8) and Eq. (7) gives Eq. (9)[59].

$$\gamma_{D \to A} = \frac{k_{D \to A}}{k_D} \tag{8}$$

$$E_{D \to A} = \frac{\gamma_{D \to A}}{1 + \gamma_{D \to A}} \tag{9}$$

In the case of one QD to one AF647 and one AF740 dye, the QD quench efficiency represents the sum of QD to AF647 and QD to AF750 FRET efficiencies, and can be calculated using Eq. (10)[59],

$$Q_{QD} = \frac{\gamma_{QD \to AF647} + \gamma_{QD \to AF750}}{1 + \gamma_{QD \to AF647} + \gamma_{QD \to AF750}} \tag{10}$$

The AF647 quenching efficiency consists of two components, competitive ($Q_{c-AF647}$) and sequential ($Q_{s-AF647}$) quenching efficiencies, and can be calculated from Eq. (11), Eq. (12), and Eq. (13)[59], respectively,

$$Q_{AF647} = 1 - (1 - Q_{c-AF647}) \times (1 - Q_{s-AF647}) \tag{11}$$

$$Q_{c-AF647} = 1 - \left( \frac{1 + \gamma_{QD \to AF647}}{1 + \gamma_{QD \to AF647} + \gamma_{QD \to AF750}} \right) \tag{12}$$

$$Q_{s-AF647} = \left( \frac{\gamma_{AF647 \to AF750}}{1 + \gamma_{AF647 \to AF750}} \right) \tag{13}$$

## Multi-exponential emission decay analysis

Emission decay curves of QDs alone and FRET-quenched QD emission were fit using a multi-exponential emission intensity decay function (Eq. (14)).

$$I = C \left[ \sum \gamma_{Dai} \exp \left( -\frac{t}{\tau_{Dai}} \right) \right] \tag{14}$$

where $C$ is the total amplitude and $\gamma_{Dai}$ are the amplitude fractions ($\sum \gamma_{Dai} = 1$) of the different FRET contributions with FRET-quenched emission decay times $\tau_{Dai}$ emission decay time averaging was performed using amplitude weighted average decay times (Eq. (15)).

$$\tau_{DA} = \sum \gamma_{Dai} \tau_{Dai} \tag{15}$$

## Molecular dynamics (MD) simulations

All MD simulations were carried out using the Large-scale Atomic Molecular Massively Parallel Simulator (LAMMPS) package[73]. The ssDNA models, poly(deoxyadenosine) (polyA) with 28–53 repeating units, were built by the BIOVIA Materials Studio software, and the consistent-valence force field was used, the parameters were directly adopted from the software[74]. The sulfur functionalized polyA was built by replacing one oxygen on the po with a sulfur and modifying the corresponding force field parameter. In the ssDNA model, the sulfur functionalized polyA part was changed to 5, 10, and 30 units, while the non-functionalized polyA part was fixed at 23 units. The ssDNA was fully ionized, each repeating unit carries −1 partial charge, and $Na^+$ ions were added to neutralize the total charge. The ZnS nanoparticle (ZnS NP) in Wurtzite crystal structure with 6 nm (5710 atoms) in diameter was built using LAMMPS, which can represent the CdSe/ZnS core/shell QD in experiments (Supplementary Fig. 52). The water environment was implicitly included in the model by setting the dielectric constant at 80. No additional buffer salt and surface ligands were included in this simple model. The long-range Coulombic force was solved by the

Ewald summation method using the particle-particle particle-mesh algorithm[75]. The force field parameters are listed in Supplementary Tables 11–16.

The simulation box size was fixed at $100 \times 100 \times 100$ nm$^3$. The total number of atoms in a typical model was between 6637 and 7462 for a 6 nm diameter ZnS NP plus one ssDNA chain. The simulation time step was 1 fs and each simulation run was 2–10 ns in total. Multiple *NVT* ensembles were applied to relax the system and acquire productive results. The Langevin thermostat with a damping constant of 100 fs was used to maintain the temperature at 300 K. During the simulation the ZnS NP was fixed at the center of the box and initially the ssDNA was set at more than 10 nm away from the ZnS NP surface. More than 50 ns of relaxing and dragging simulation processes were used to prepare for the representative structure of one ssDNA wrapped on the ZnS NP (Supplementary Movie 2). Typical procedures were written in Supplementary Information. Then the productive simulation was carried for 2 ns to calculate the distance between the free head of the ssDNA to the ZnS NP surface and the radius of gyration (*Rg*) of the ssDNA. At the end of the 2 ns productive simulation, the effective contact area between the ssDNA (23 non-functionalized poly A units) and the ZnS NP was calculated by the Voronoi tessellation method[76].

## Data availability
The data generated in this study, including all the parameters used to model the data, are provided in the article, Supplementary Information and Source Data file. Source data are provided with this paper.

## Code availability
Open Source Software to generate wireframe DNA origami structures is available at https://github.com/lcbb/athena

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

## Acknowledgements

The authors are grateful to support from NSF OAC-1940231 and NSF CBET-1729397 to C.C., J.L.B., and M.B., from NSF OAC-1940152 to X.W., Y.Z., and R.H., from NSF CHE-1839155 to C.C., J.L.B., G.S.S.-C., and M.B., from ONR N00014-20-1-2084 and NDSEG Fellowship to M.F.P., and from NIH Director's New Innovator Award 1DP2GM128200-01 to G.S.S.-C. This research was sponsored by the U.S. Army Research Office and accomplished under cooperative agreement W911NF-19-2-0026 for the Institute for Collaborative Biotechnologies. The computing resources necessary for the molecular dynamics simulations were provided in part by the National Science Foundation through XSEDE resources under grant number CTS090079, and the Advanced Research Computing at Hopkins (ARCH) facilities supported by an NSF MRI Grant (OAC-1920103).

## Author contributions

C.C. conceived the valence-geocoded QD, designed and performed QD phase transfer and surface modification, ssDNA wrapping and wireframe DNA origami wrapping hypothesis experiment, QD-dye FRET absorbance and steady-state measurement, wireframe DNA

origami objects folding and purification, synthesis of QD-based target assembly (QD-dye and QD-QD), AGE and TEM characterization for each step, and drafted the paper. X.W. and Y.Z. developed the MD simulation model. M.F.P. designed the pentagonal pyramid wire-frame DNA origami objects. J.G. and M.N.S. measured and analyzed the time-resolved decay curves of QD-dye FRET. G.S.S.-C. supervised the time-resolved decay curves measurement. R.H. supervised the MD simulations. M.B. conceived of the QD-DNA origami assemblies and supervised the entire project. C.C., X.W., J.L.B., and M.B. wrote the paper. All authors discussed the results and commented on the paper.

## Competing interests

J.L.B. and M.B. are co-founders and equity holders of Cache DNA, Inc., a company that is commercializing the application of DNA nanotechnology for data storage. J.L.B. is an employee of Cache DNA, Inc. The remaining authors declare no competing interests.
