## [Peer Review File · Nature Communications]

nature portfolio

Peer Review File

Draft OnlyREVIEWER COMMENTS

Reviewer #1 (Remarks to the Author):

This manuscript showcases the conjugation of quantum dots (colloidal nanoparticles) using "chimeric" DNA strands that in these experiments were composed of tracts of single-stranded DNA with a phosphorothioate backbone followed by sequence with phosphate backbone of varying length. Authors followed the conjugation process in silico with molecular dynamic simulations. Finally, DNA-covered quantum dots were bound to DNA origami structures.

This article is certainly the result of a lot of work and it does contain useful information for people working in the field. However, I think it lacks the type of breakthrough or fundamental result that Nature Communications potentially strives for. Authors make the point that their method would enable them "to fabricate hybrid materials for functional nanoscale photonic devices". Unfortunately, the manuscript falls short of demonstrating this, e.g. a spatially controlled cascade of quantum dots yielding designed optical properties. I would recommend acceptance only if the authors could show such (or any other) meaningful application of their conjugation method in a revised version of the manuscript. At the current stage, an applied chemistry journal could be suitable outlet for this work.

- Quantum dots of many types have been conjugated in many ways with DNA. In particular, the use of tracts of phosphorothioate backbones is well established (doi: 10.1021/acsami.9b07717; a recent review by Dubertret et al.: "Quantum dots–DNA bioconjugates: synthesis to applications"; doi:10.1098/rsfs.2016.0064)

- Monovalent quantum dots have been produced in multiple ways (e.g. doi: 10.1038/nmeth.2682 or doi:10.1021/bc5002032 or doi:10.1002/cbic.200900300, doi: 10.1016/j.biomaterials.2011.04.013 etc.)

- Quantum dots have been placed on DNA origami in several ways: with streptavidin (e.g. doi: 10.1021/nl101079u, doi: 10.1002/adfm.201102077) and with phosphorothioated DNA (doi: 10.3390/nano9030339).

Other comments:

- I had trouble finding the composition of the quantum dots used. It is not stated in the text. Is it CdSe/ZnS? I did a web search with the information given in the materials section ... or did I overlook it somewhere?

- Fig 2 shows FRET efficiencies in various configurations of dyes and quantum dots in bulk. As the authors make a point of monovalent functionalization, single-molecule FRET would be a way to prove this.

- Concerning the same figure / data: did the authors perform experiments with the dye in a distal configuration? Figure 2a displays only proximal configurations. This way additional information could be retrieved.

Reviewer #2 (Remarks to the Author):

The authors present a novel approach to modify QDs with controlled number and arrangement of surface functionalities. This is a critical challenge of QDs, and particularly DNA-functionalized QD assemblies, to enable constructs for a wide range of applications such as imaging probes, biosensors, and photonic devices and materials. The authors use a previously demonstrated approach of wrapping of chimeric ssDNA strands where part of the strand is composed of a phosphorothioate (ps) backbone and the other part is composed of a normal phosphate backbone. They integrate fabrication, experiments, and computational modeling to study the wrapping process to understand parameters that lead to wrapping of a single chimeric ssDNA with an accessible po binding site for functionalization. They expand this approach using stiff (2HB or 6HB edges) wireframe DNA origami structures, specifically a tetrahedron appropriate for small QDs and pentagonal pyramid structure appropriate for larger QDs. These DNA origami structures leverage the ssDNA wrapping to achieve a single tetrahedron with high efficiency or a single pentagonal pyramid with good efficiency on QDs. Then the authors leverage the site addressability of DNA origami to demonstrate effective assembly of QD fluorophore assemblies or multi-QD assemblies. This work expands on prior studies for QD and more broadly nanoparticle functionalization and adds important mechanistic studies that can inform materials design and methods for controlling valency and geometry of QD functionalization/assembly. The results are impressive, and I think valuable to the community. However, there are several points that should be addressed and/or clarified. In particular, the authors talk about this as a general strategy and they point to limitations of previous methods like specific to QD size or other parameters, inability to efficiently control valency, or limitations with respect to adding functionalities very near the surface. I agree those are limitations, and I agree the methods presented here provide valuable improvements, however there are still important limitations of the current methods. For example, for the larger QDs the efficiency of monovalency was relatively low (~70% for the Pep-QD660, which are ~14nm QDs). Also, the authors mention limitations of bulky streptavidin approaches adding space between the QD and DNA construct of ~5nm of the surface, but here they only show adding functionalities at ~7nm from the surface. Is

there a reason fluorophores were not added nearer to the surface? Overall, I think there is valuable work but the authors should clearly present the specific advances and also present the limitations (or areas of future improvement) of their approach. I the claiming of a “general framework” needs to be clarified, since only specific cases are demonstrated and some are rather limited in assembly efficiency (e.g. 70% efficiency for the Pep-QD660 assembly). And in some cases assembly efficiencies were not provided. There are also a number of specific points below that should be addressed to improve the clarity of the results and methods.

- Line 71: The sentence that starts, “In contrast to tile-based self-assembly....” Is unclear. I think it is just worded awkwardly.

- Line 78: What is meant by “quantitative fabrication yield”? Do the authors mean the yield is 100%? This is a minor point, but I would not claim 100% yield unless it is fully demonstrated, and typical characterizations, like gel electrophoresis are not sufficient to make the claim of 100% perfectly folded structures.

- Figure 2b: I find it difficult to see everything in this plot. I would suggest showing in separate plots the for the QD600, QD630 and AF657 and another for the QD660 and AF750. Also I do not think the “PL” in the Normalized “PL” axis is defined anywhere. Also the colors of letting in the legend is a bit different than the curves making it difficult to map everything together.

- Lines 196-197: I do not understand the comment, “which may be due to a polydisperse set of conformations of the chimeric ssDNA when wrapped on the QD surface.” To me, a polydisperse set of conformations would have to do with either multiple bands or the thickness of the band, not necessarily a minor shift relative to the control or to other strand lengths.

- Lines 199-201: It does not look to me like the mobility of the QD wrapped with 0 nt A* is similar to the monovalent band of the thiolated-ssDNA-wrapped QD600. The monovalent band of the thiolated-ssDNA-wrapped QD600 runs faster, more like the 5 nt A* or 10 nt A* wrapped QDs. Also, I am confused about claim about the non-specific adsorption. To me the shift of the 0 nt A* relative to the bare QD600 would suggest non-specific adsorption of the 0 nt. And I do not think the fact that the mobilities of the thiolated-ssDNA-wrapped QD600s are similar to any of the A* wrapped QDs would necessarily suggest non-specific surface adsorption of the thiolated QDs. I am not sure if that is what the authors are suggesting. The experiment described later in the paragraph clarifies that there is non-specific adsorption of the AF647 po strand. But the authors should clarify the description of the gel results here.

- Lines 247-248: How was the mean donor(QD)-acceptor(AF750) distance calculated from TEM? Figure S2 only shows diameter measurements.

- Lines 267-274: It took me awhile to understand what was being shown in these gels. I believe the green signal in in Fig. 3d shows QDs and DNA nanostructures while the yellow only shows QDs. I think it would be helpful to specifically state the emission spectra of the SYBR Safe and the QD600 overlap in the main text description. Also the later description talks about QD600 (Line 273) with respect to Figure 3i, but the 3i shows QD660s (I think that was just a typo). It would also be helpful

to label in the gel image or in the figure caption (or both) what colors show what in the two gels in figure 3.

- Line 284: What was the yield of the properly assembled Tet-QD600 assembly? The gel appears to show a leading edge that may only be DNA structure, and it seems there are a few (very few) DNA origami Tets without QDs in figure 3c. My impression is the yield is quite good, but it would be good to quantify.
- Lines 309-311: Were TEM images taken of these Tet-QD constructs? Can the authors quantify yield? It appears to be quite good from the gel, but TEM images would be useful to confirm.
- Figure 4C: how were these constructs designed? It seems there is one wrapped chimeric DNA that binds 1-4 AF647. I do not see how that can be done but maintain all of the dyes at ~7nm. Also, how is the 7nm spacing for this construct determined? Is it just based on the length of the duplex? The figure is somewhat confusing also because it shows different color dots as the dyes, are those all AF647? And they are anchored at different positions, but I understood there is only one strand wrapped?
- Line 332: What is the yield of these? The yields look very good to me, but it would strengthen the results to give a quantitative number, which could be estimated from TEM images, instead of just saying “high fidelity”.
- Lines 336-339: How are these angles designed in the DNA complex-based QD trimers?

Reviewer #3 (Remarks to the Author):

The work by Chen et al reports a use of DNA frames for controlling positions of quantum dots. Although the topic is of broad interest for self-assembly, as for applications in nano-optics, imaging and light manipulation, there are various important aspects that must be addressed.

First general takeaway is that the authors are making claims that go far beyond this work – they aren't the first to use these types of chimeric DNA strands to mono-functionalize quantum dots, nor are they the first to attach materials to DNA origami structures (which they note). The authors seem to be confirming that these bind to overhangs on a DNA origami structure – there are a lot of experiments here, but this seems to be the main point, in addition to claiming that this strategy is more effective than other approaches.

Second takeaway – given the conclusion of this work that the DNA wrapping path towards mono-functionalization is better than more traditional methods of DNA attachment to quantum dots, both because of the valence and because of the bulkiness of certain functional groups such as biotin-streptavidin. The paper even goes through measurements to get these actual values for spacing, distance, some function for a FRET dye network. However, none of this compared to any of the systems the authors are claiming need to be improved upon. There is need for control experiments and comparisons, otherwise these claims are not supported. The authors use of a thiolated DNA sequence earlier in the paper to show that the thiolated sequence leads to multiple attached DNA strands – they felt they should show a control here with a more traditional ssDNA (though don't understand the need for the thiol group as it doesn't appear to be used). However, this experimental "control" group is missing everywhere else in the paper.

As a schematic diagram Figure 1 does a good job outlining the method that the authors used in this study. As for the results shown in Figure 2, I either cannot see major differences to the use of chimeric DNA to previous results by other groups or the authors did not convey those points clearly enough in the writing. The authors of this study even cite previous work using this method, but use Figure 2 to essentially reaffirm this method, unrelated to the DNA origami design at all.

The paper seems unclear what its main points are. Much of the beginning half of the paper is dedicated to work that has, in large part, been previously published. All of this is supposedly done to build a picture of control for binding to origami structure to enable functionality not truly achievable with other methods of functionalizing quantum dots. Yet, while Figure 3 and 4 do get into binding to origami, there is never a comparison of structural yields for binding or functionality of the quantum dots in comparison to other systems. The discussion then seems to turn to the rigidity of the origami structure – which again, is already known, and has been shown many times before. Effectively, the story seems to change in the middle.

In Figure 3, the particles are semi-surrounded by the structure they are binding to, thus negating the argument that mono-functionalization leads to reduced aggregation of structure because otherwise the particles could bind other structures if they have multiple binding sequences. However, the specific structures the authors tested would cloud this conclusion in the first place – it is not clear why a multi-valent quantum dot would give any different results in the structures shown in Figure 3, given that the structure most effectively binds one quantum dot, and then sterically hinders the quantum dot from binding to other structures.

Figure 3 focuses on the localization of QDs to the DNA origami. The use of gel and TEM here does a good job to show that the QDs become fixed to their target site, however this figure would be a good place to show statistics. The authors show in Supplementary Fig. 24 that to achieve a 70% yield of correctly placed monovalent QDs, they needed to use 4:1 molar excess, otherwise a 1:1 would result in a 52% loading efficiency. So, I am still not seeing the real need for chimeric DNA.

Figure 4, as stated previously, seems to lose the story. The control used for a comparison here is a floppy DNA linking structure, as opposed to the structural DNA origami system. The point of the paper wasn't that DNA origami can organize materials more rigidly and effectively than a floppy DNA structure, it was the quantum dot functionalization technique yielded differences in quantum dot function and organizational yield when compared to other techniques. Neither are shown.

My general thought with this paper is that they initially convey to the reader that there are major issues with the more conventional methods of QD localization (biotin-streptavidin) when it comes to FRET probes or networks and the possibility of the same QD being attached to two separate sites. However, there is no control experiments, either in the assembly of the QDs to the origami or measuring the FRET efficiency where the biotin-streptavidin method was used. By claiming that this method resolves these issues but by not providing any qualitative or quantitative benchmarks, the reader is left unclear as to how much this method improves results compared to these others.

I don't see how the paper could be published without additional experiments and clarifying the claims.

Some other comments:

A* has a shifting definition through the paper, sometimes referring to just Ps and sometimes referring to Ps + Po. For example, lines 152-154 say the A* contains the Po sequence, but otherwise does not seem like it does.

The "floppy" DNA complex-QD-dye system did not seem that much worse than the rigid structure – can the authors define what they are defining as significant? In multiple setups, the FRET differences don't appear to be statistically significant. Also, "Floppy" structures in Figure 4 - what is this structure? Is there a different design? What is the DNA complex-QD-dye system – Figure 4I isn't enough to describe this. There isn't sufficient explanation here how are formed in the first place. Are these formed through the mechanism in Reference 18?

Reviewer #4 (Remarks to the Author):

The manuscript "Nanoscale 3D Spatial Addressing and Valence Control of Quantum Dots using Wireframe DNA Origami" describes a framework for attaching quantum dots (QDs) to addressable sites on DNA origami. The use of origami provides precise control over the positioning of QDs and ordinary fluorophores, which has obvious advantages for sensing applications, but in principle could have impacts on technologies ranging from quantum computing to light bulbs. The work is timely and, overall, well presented, though I have a few concerns about the simulation portion of the study. Hence, I recommend the manuscript for publication provided the authors address the specific concerns raised below.

Specific Concerns:

1. It's not immediately clear what the consistent-valence force field used for the simulations actually is. The only reference listed is for the commercial BIOVIA package; did the BIOVIA team create this force field? The authors included the force field parameters in the SI, but it's not clear how they would map onto the ssDNA atoms, at least without access to the BIOVIA package. If a reference for the force field is unavailable, it would be helpful to cite other papers using the force field to model similar systems.
2. The simulations should be longer. A five nanosecond

simulation isn't long by contemporary standards even when explicit solvent is used (AMBER, for example, can simulate hundreds of nanoseconds/day). Single-stranded DNA is a very flexible molecule, so quantities like the radius of gyration or the persistence length should be averaged over a large ensemble of conformations. The authors would ideally extend the simulations and provide evidence that averages are converged.

3. Supporting animations of the MD simulations may give a sense of the fluctuations and dynamics of the DNA.

Draft Only

REVIEWER COMMENTS

Reviewer #1 (Remarks to the Author):

This manuscript showcases the conjugation of quantum dots (colloidal nanoparticles) using "chimeric" DNA strands that in these experiments were composed of tracts of single-stranded DNA with a phosphorothioate backbone followed by sequence with phosphate backbone of varying length. Authors followed the conjugation process in silico with molecular dynamic simulations. Finally, DNA-covered quantum dots were bound to DNA origami structures.

This article is certainly the result of a lot of work and it does contain useful information for people working in the field. However, I think it lacks the type of breakthrough or fundamental result that Nature Communications potentially strives for. Authors make the point that their method would enable them "to fabricate hybrid materials for functional nanoscale photonic devices". Unfortunately, the manuscript falls short of demonstrating this, e.g. a spatially controlled cascade of quantum dots yielding designed optical properties. I would recommend acceptance only if the authors could show such (or any other) meaningful application of their conjugation method in a revised version of the manuscript. At the current stage, an applied chemistry journal could be suitable outlet for this work.

Response: We appreciate the positive comments and perspective of the reviewer. The principal breakthrough of this article is a generalized method to precisely control the position and stoichiometry of multiple QDs on 3D wireframe DNA origami scaffolds. While numerous strategies have been developed to control the positions of QDs on DNA origami, they either lack the ability to control QD valences or they introduce bulky conjugates, such as the often-used streptavidin-biotin conjugation linkers. This latter approach is a critical limitation of existing strategies for some applications, for example FRET, catalysis, and sensing, which all benefit from sub-nanometer control of 3D spatial positioning. And unlike 3D bricklike origami that are typically dense bundles of parallel duplexes, meshlike 3D wireframe origami facilitate 3D positioning of QDs that are otherwise challenging to achieve. Towards this end, we demonstrated the use of a DNA wrapping strategy to control QD valency by spatially positioning individual QDs on 3D wireframe DNA origami scaffolds. This DNA wrapping strategy excludes the use of bulky linkers, which allowed us to precisely control the spatial organization of multiple QDs and dyes.

In our revised manuscript, to better demonstrate the advantages of this strategy over existing approaches, we designed and fabricated three types of concentric multi-step FRET networks based on the 3D spatial addressing and valence control of QDs. FRET provides a very sensitive readout of fluorophore distances, enabling us to compare quantitatively our approach with the use of simpler streptavidin-biotin linkers. We showed three types of concentric multi-step FRET networks can be achieved with fixed stoichiometry (1:1:1 in this work) between fluorophores, due to the combined 3D spatial addressing and valence control of QDs using our approach (Figs. 5e–5g). In each case, we demonstrated that FRET measurements are in agreement with theoretical FRET values anticipated from rigidly constrained fluorophores that do not use bulky linkers. We discuss these results on pages 20–22 of our revised manuscript, and show the primary referenced Figure below.

Fig. 5. Valence-geocoded QD-based energy transfer circuits. (a) Schematic of the Tet-QD-based FRET network and the overhang design (the distance for each de-excitation pathway is ~ 7 nm). (b) AGE (1.5%) image for 1-4 AF647-labeled Tet wireframe DNA origami objects. (Green: SYBR safe channel, Red: AF647 channel). (c) Schematic of the DNA complex-QD-based FRET network. (d) FRET efficiencies as a function of acceptors calculated from the Tet-QD-dye system, the DNA complex-QD-dye system, and Förster theory (Eq. 5). P -values are from Student's t -test ($*P < 0.05$); all P -values are provided in Supplementary Table 3. (e) Schematics of three types of Pep-QD-based multi-step FRET networks and predictions of distances between initial donor (QD), relay (AF647), and acceptor (AF750). (f) QD quenching efficiencies and (g) AF647 quenching efficiencies calculated from the Pep-QD based multi-step FRET networks according (darker bar), and Förster theory (lighter bar) (Eqs. 6-13). Error bars in Figs. 5d, 5f, and 5g represent standard deviations of the mean ($n = 3$ replicates per group).

- Quantum dots of many types have been conjugated in many ways with DNA. In particular, the use of tracts of phosphorothioate backbones is well established (doi: 10.1021/acsami.9b07717; a recent review by Dubertret et al.: "Quantum dots–DNA bioconjugates: synthesis to applications"; doi:10.1098/rsfs.2016.0064)

- Monovalent quantum dots have been produced in multiple ways (e.g. doi: 10.1038/nmeth.2682 or doi:10.1021/bc5002032 or doi:10.1002/cbic.200900300, doi: 10.1016/j.biomaterials.2011.04.013 etc.)

- Quantum dots have been placed on DNA origami in several ways: with streptavidin (e.g. doi: 10.1021/nl101079u, doi: 10.1002/adfm.201102077) and with phosphorothioated DNA (doi: 10.3390/nano9030339).

Response: We appreciate the point of the reviewer. QDs have indeed been conjugated in many ways using DNA, and these approaches can generally be divided into two categories: Affinity for existing conjugated ligands and affinity for the cationic shell of QDs (*Interface Focus*. 2016 Dec 6;6(6):20160064). Monovalent

QDs have also been produced in multiple ways, including steric exclusion (*Nat. Methods* **2013**, *10*, 1203–1205), engineering a streptavidin tetramer with only one functional biotin binding subunit (*Nature Methods*, **2006**, *3*, 267–273), purification by ion exchange using diethylaminoethyl (DEAE) sepharose-packed spin columns (*ChemBioChem* **2009**, *10*, 1781–1783) or using DEAE functionalized magnetic beads (*Bioconjugate Chem.* **2014**, *25*, 1342–1350). However, all of these preceding methods failed to achieve the 3D spatial addressing and valence control of QDs, which is the main point of our paper.

DNA nanotechnology has been used to organize individual QDs in 1D, 2D, and 3D systems, including organization of QD-DNA conjugates by complementary base pairing to DNA origami structures (*J. Am. Chem. Soc.* **2012**, *134*, 17424–17427, *Nat. Nanotechnol.* **2014**, *9*, 74–78), using QDs with a peptide nucleic acid (PNA) containing a hexahistidine peptide motif (*ACS Nano* **2021**, *15*, 9101–9110), using streptavidin-biotin conjugation (*Nano Lett.* **2010**, *10*, 3367–3372), and incorporating phosphorothioated DNA during ZnS layer growth (*Nanomaterials* **2019**, *9*, 1–10). However, the preceding strategies still suffer from multiple ssDNAs, PNAs, or phosphorothioated DNA that are typically bound non-specifically to the QD surface, which increase the probability that a QD will bind to more than one DNA nanostructure, which consequently reduces the purity of target assemblies. In particular, for the fabrication of valence-geocoded QDs, the QD and DNA frame need to be of similar size. As shown in Fig. 3, in the case of biotin-streptavidin conjugation, streptavidin-functionalized QDs that contain multiple valences (active streptavidins) can bind one (monovalent) or two (divalent) Pep with biotin-modified overhangs at the inner center, and up to three (trivalent) Pep with biotin modified overhang at the outer edge. Thus, controlling the ssDNA valence of individual QDs is crucial to avoid multiple DNA origami objects attaching to a single QD.

Moreover, the bulky biotin-streptavidin complex separates the QD surface from the wireframe DNA origami object by at least 5 nm, impeding the implementation of efficient Förster resonant energy transfer (FRET)-based QD probes or FRET networks for biosensing and imaging, molecular logic and computing (Supplementary Figs. 42 and 45). In this paper, we demonstrated that wireframe DNA origami objects and nanoparticle surface modifications can translate nanoscale DNA origami design strategies into functional hybrid nanoparticle materials. And the chimeric ssDNA wrapping approach can be applied to the overhang of a DNA origami structure, which can achieve the mono-functionalized QD-DNA origami assemblies with 3D spatially addressable valence, also minimizing the distance between the QD and wireframe DNA origami object itself for efficient energy transfer.

Fig. 3. Wireframe DNA origami-based chimeric ssDNA wrapping and biotin-streptavidin conjugation. (a) Schematic and TEM images of Pep-QD assemblies using chimeric ssDNA (30 nt A*) wrapping or biotin-streptavidin conjugation at the inner center and outer edge of Pep wireframe origami objects. The percentage of mono-, di, and tri-valent Pep-QD assemblies were calculated from TEM images (308 and 337 assemblies for Pep with 30 nt A* domain at the inner center and outer edge, respectively; 257 and 265 assemblies for Pep with biotin domain at the inner center and outer edge, respectively). (b) AGE (0.8%) image (four-fold excess Pep) of QD630 alone, Pep-30 nt A*-QD630 (inner center), Pep-30 nt A*-QD630 (outer edge), Pep-30 nt A*(inner center), and Pep-30 nt A* (outer edge). (c) AGE (0.8%) image (four-fold excess QD) of streptavidin QD660 alone, Pep-biotin-streptavidin QD655 (inner center), Pep-biotin-streptavidin QD655 (outer edge), Pep-biotin (inner center), and Pep-biotin (outer edge). Representative TEM images of (d) Pep-30 nt A*-QD630 (inner center), (e) Pep-30 nt A*-QD630 (outer edge), (f) Pep-biotin-streptavidin QD655 (inner center), and (g) Pep-biotin-streptavidin QD655 (outer edge). (Scale bars: 20 nm (a); 50 nm (d), (e), (f), and (g).)

Supplementary Fig. 42. a) Schematic of Pep-30 nt A* QD600-AF647 (left), Pep-30 nt A* QD630-AF647 (middle), and Pep-biotin-streptavidin QD605-AF647 (right). Representative PL spectra of b) Pep-30 nt A* QD600-AF647, c) Pep-30 nt A* QD630-AF647, and d) Pep-biotin-streptavidin QD605-AF647. e) AF647 to QD surface distances in Pep-30 nt A* QD600-AF647 (red), Pep-30 nt A* QD630-AF647 (blue), and Pep-biotin-streptavidin QD605-AF647 (green). The distances were determined by donor-acceptor distance (Equation 5) and radius of QD600, QD630, and streptavidin QD605 (~3.2 nm, ~4.0 nm, and ~3.8 nm, Supplementary Figs. 2 and 16).

Supplementary Fig. 45. a) Schematic of Pep-30 nt A*-QD600-AF647-AF750 and Pep-biotin-streptavidin QD605-AF647-AF750-based multi-step FRET networks. Representative PL spectra (top) and zoomed area of b) Pep-30 nt A*-QD600-AF647-AF750-based multi-step FRET networks and c) Pep-biotin-streptavidin QD605-AF647-AF750-based multi-step FRET networks.

Other comments:

- I had trouble finding the composition of the quantum dots used. It is not stated in the text. Is it CdSe/ZnS? I did a web search with the information given in the materials section ... or did I overlook it somewhere?

Response: They are CdSe/ZnS quantum dots, which was not stated in the manuscript. We added this information to Method-General materials: CdSe/ZnS core/shell quantum dots (catalog number: 900218(QD600), 900220(QD630) and 900249(QD660))

- Fig 2 shows FRET efficiencies in various configurations of dyes and quantum dots in bulk. As the authors make a point of monovalent functionalization, single-molecule FRET would be a way to prove this.

Response: We appreciate the Reviewer's suggestion. While single-molecule FRET data could certainly provide further support of monovalent functionalization, we instead fabricated the QD600 dimer using monovalent QD600 and confirmed correct structural formation directly with TEM in Supplementary Fig. 13. Moreover, TEM images of Pep-QD630 assemblies using ps-backbone wrapping domain also demonstrated the monovalent functionalization (Fig. 3), and the divalent yield was less than 1% for the ps-backbone based overhang at the inner center or outer edge (Supplementary Fig. 24).

Supplementary Fig. 13. a) Schematic of preparation for QD dimers using monovalent QDs. b) The yield of QD600 dimers calculated from TEM images (166 NPs). c) TEM images of QD dimer at several magnifications (Scale bar: 20 nm).

Supplementary Fig. 24. The yield of Pep-30 nt A*-QD630 assemblies using ps-backbone wrapping domain at the a) inner center and b) outer edge. The yield of Pep-biotin-streptavidin QD655 assemblies using biotin domain at the c) inner center and d) outer edge. Yields were calculated from TEM images (308 assemblies for a, 337 assemblies for b 257 assemblies for c), and 265 assemblies for d)).

- Concerning the same figure / data: did the authors perform experiments with the dye in a distal configuration? Figure 2a displays only proximal configurations. This way additional information could be retrieved.

Response: Thank you for pointing out this issue. In the case of QD-AF647 FRET in a distal configuration, the FRET efficiencies were $10 \pm 2\%$ and $13 \pm 2\%$, and donor-acceptor distances increased to 8.3 ± 0.3 and 8.8 ± 0.2 nm (mean \pm standard deviation; $n = 3$) for QD600-distal-AF647 and QD630-distal AF647 FRET pairs, respectively, which were consistent with the distance ranges calculated from potential QD-DNA duplex-dye geometries (Supplementary Fig. 8). For QD660-AF750 FRET in a distal configuration, there is no FRET since the distance (10.2-14.4 nm) between the QD center and AF750 is nearly 2x the Förster distance (10.6 nm, Supplementary Table 2).

We have discussed the above-mentioned result in the revised manuscript on page 10.

Supplementary Fig. 8. a) Schematic of ssDNA containing 30 nt A* and fixed po domain (21 nt) wrapping on QD600 and QD630, and hybridization with AF647-labelled complementary strand. Photoluminescence (PL) spectra of b) QD600-distal-AF647 and c) QD630-distal-AF647 FRET pairs. FRET efficiencies were $10 \pm 2\%$ and $13 \pm 2\%$, and donor-acceptor distances increased to 8.3 ± 0.3 and 8.8 ± 0.2 nm (mean \pm standard deviation; $n = 3$) for QD600-distal-AF647 and QD630-distal AF647 FRET pairs, respectively. d) Possible structures of DNA duplex wrapping on the QD and corresponding donor-acceptor distances (7.8-10.4 nm and 8.2-11.2 nm for QD600-distal-AF647 and QD630-distal-AF647, respectively).

Reviewer #2 (Remarks to the Author):

The authors present a novel approach to modify QDs with controlled number and arrangement of surface functionalities. This is a critical challenge of QDs, and particularly DNA-functionalized QD assemblies, to enable constructs for a wide range of applications such as imaging probes, biosensors, and photonic devices and materials. The authors use a previously demonstrated approach of wrapping of chimeric ssDNA strands where part of the strand is composed of a phosphorothioate (ps) backbone and the other part is composed of a normal phosphate backbone. They integrate fabrication, experiments, and computational modeling to study the wrapping process to understand parameters that lead to wrapping of a single chimeric ssDNA with an accessible binding site for functionalization. They expand this approach using stiff (2HB or 6HB edges) wireframe DNA origami structures, specifically a tetrahedron appropriate for small QDs and pentagonal pyramid structure appropriate for larger QDs.

These DNA origami structures leverage the ssDNA wrapping to achieve a single tetrahedron with high efficiency or a single pentagonal pyramid with good efficiency on QDs. Then the authors leverage the site addressability of DNA origami to demonstrate effective assembly of QD fluorophore assemblies or multi-QD assemblies. This work expands on prior studies for QD and more broadly nanoparticle functionalization and adds important mechanistic studies that can inform materials design and methods for controlling valency and geometry of QD functionalization/assembly. The results are impressive, and I think valuable to the community. However, there are several points that should be addressed and/or clarified. In particular, the authors talk about this as a general strategy and they point to limitations of previous methods like specific to QD size or other parameters, inability to efficiently control valency, or limitations with respect to adding functionalities very near the surface. I agree those are limitations, and I agree the methods presented here provide valuable improvements, however there are still important limitations of the current methods. For example, for the larger QDs the efficiency of monovalency was relatively low (~70% for the Pep-QD660, which are ~14nm QDs). Also, the authors mention limitations of bulky streptavidin approaches adding space between the QD and DNA construct of ~5nm of the surface, but here they only show adding functionalities at ~7nm from the surface. Is there a reason fluorophores were not added nearer to the surface? Overall, I think there is valuable work but the authors should clearly present the specific advances and also present the limitations (or areas of future improvement) of their approach. I the claiming of a “general framework” needs to be clarified, since only specific cases are demonstrated and some are rather limited in assembly efficiency (e.g. 70% efficiency for the Pep-QD660 assembly). And in some cases assembly efficiencies were not provided. There are also a number of specific points below that should be addressed to improve the clarity of the results and methods.

Response: We appreciate the reviewer pointing out this issue, which is important due to three reasons. First, we should correct the way we calculated the efficiency of monovalency for Pep-QD660. Second, we have not explained well the way in which we calculated the donor-acceptor distance in the Tet-QD-dye FRET pair in the manuscript. Finally, we should clearly present both the advances and limitations of our approach.

First, the ~70% yield of monovalency for Pep-QD660 was calculated with large clusters (X in the histogram of Supplementary Fig. 24 earlier version) in an earlier version of our manuscript. In other literature, large clusters were not counted because they were likely due to co-localization or aggregation of clusters during TEM sample preparation (*J. Am. Chem. Soc.* **137**, 4320–4323 (2015)). The efficiency of monovalency was up to 84% for Pep-QD660 when we did not count the large clusters (Supplementary Fig. 36). And the yields of Pep-QD630 and Tet-QD600 were up to 85% and 91%, respectively. We claim a “general framework” because our method can in principle be applied to any wireframe DNA origami object with various geometries to achieve different 3D spatial addressing of QDs. And the design of a library of wireframe DNA origami can easily be achieved by our automated sequence design algorithm for wireframe scaffolded DNA origami (ATHENA).

Second, the reason why we designed a ~7 nm distance between QD and dye is that we sought to compare FRET from our wireframe origami to DNA complexes at the same donor-acceptor distance. Considering the radius of QD600 (~3.2 nm), the distance between functionalities and QD surface should be ~4 nm. The distances between adding conjugates and the QD surface also depend on the wireframe origami geometries. For Pep-QD assemblies, the AF647 was labeled close to the vertex of the Pep frame. Distances between AF647 and the surface of the three types of QD (QD600, QD630, and streptavidin-QD605) were 2.8 nm, 3.1 nm, and 7.4 nm, respectively, using QD-AF647 FRET calculations (Supplementary Fig. 42). Streptavidin-QD605 were measured as being further from the AF647 due to the spacing introduced by the bulky streptavidin tags, leading to reduced FRET efficiencies.

Finally, to present the specific advances of our approach, we added a series of experiments to compare the chimeric ssDNA wrapping and biotin-streptavidin conjugation approaches for QD-DNA origami assemblies in our revised manuscript. To test our ability to bind a single wireframe DNA origami within a QD, pentagonal pyramid (Pep) wireframe origami objects with an overhang containing ps-backbone (30 nt A*) at the inner center or outer edge were used to fabricate Pep-30 nt A*-QD630 assemblies. In contrast, Pep with a biotin-modified overhang at the inner center or outer edge were used to fabricate Pep-biotin-streptavidin-QD655 assemblies (Fig. 3a). QD630 and streptavidin-QD655 were used to fabricate the assemblies since they had similar sizes and were easily observable using negative-stain TEM. AGE gel shift (Figs. 3b, 3c and Supplementary Figs. 18 and 19) and negative-stain TEM imaging (Figs. 3a, and 3d-3g, Supplementary Figs. 20-23) of the Pep-QD assemblies validated the assembly of the target DNA origami objects and different stoichiometric ratios of Pep per QD using chimeric ssDNA wrapping and streptavidin-biotin conjugation. By using chimeric ssDNA wrapping, each QD can only bind with a single Pep wireframe DNA origami object with an overhang at the inner center or outer edge. However, in the case of biotin-streptavidin conjugation, a streptavidin QD that contains multiple valences (active streptavidins) can bind one (monovalent) or two (divalent) Pep with biotin-modified overhang at the inner center, and up to three (trivalent) Pep with a biotin modified overhang at the outer edge. Although the loading yield (Pep-QD/Pep) of Pep-30 nt A*-QD630 (inner center: 85%; outer edge: 87%) and Pep-biotin-streptavidin QD655 (inner center: 90%; outer edge: 86%) were similar, the yields of the correct assemblies (monovalent Pep-QD/Pep) were nearly equal to the loading yields for the chimeric ssDNA wrapping strategy, but only 68% for biotin-streptavidin conjugation at the inner center, and 61% for biotin-streptavidin conjugation at the outer edge (Supplementary Fig. 24). We noticed that a high proportion of di- and tri-valent Pep-QD assemblies could still form even when the Pep were incubated with four-fold excess streptavidin QD655 (Figs 3c, 3f and 3g), which may be due to the high-affinity biotin-streptavidin interaction ($K_d = 10^{-14}$ M). This result further demonstrated that the monovalent ssDNA wrapping is crucial for efficient and correct QD-DNA origami-based nanofabrication.

In the current stage, we only fabricated QD-QD assemblies using a single type of QD since we used the wireframe DNA origami objects with ps-backbone overhang to wrap the QD directly. This limitation of our approach can be addressed by using monovalent QD with different staple overhangs to involve DNA origami folding. We would like to improve our approach in this direction in future work.

We have discussed the above-mentioned result in the revised manuscript on pages 14-16, 18-19, 21-22, and 24.

Supplementary Fig. 36. a) Schematic for preparation of Pep-QD660 assemblies using 1:1 and 4:1 initial molar ratio incubation. b) The yield of Pep-QD660 assemblies calculated from TEM images (376 NPs for 1:1 (top), 261 NPs for 4:1 (bottom). Large clusters were not counted because they were likely due to co-localization or aggregation of clusters during TEM sample preparation). c) Representative TEM images of Pep-QD660 assemblies with 1:1 (top) and 4:1 (bottom) initial molar ratio.

Supplementary Fig. 42. a) Schematic of Pep-30 nt A* QD600-AF647 (left), Pep-30 nt A* QD630-AF647 (middle), and Pep-biotin-streptavidin QD605-AF647 (right). Representative PL spectra of b) Pep-30 nt A* QD600-AF647, c) Pep-30 nt A* QD630-AF647, and d) Pep-biotin-streptavidin QD605-AF647. e) AF647 to QD surface distances in Pep-30 nt A* QD600-AF647 (red), Pep-30 nt A* QD630-AF647 (blue), and Pep-biotin-streptavidin QD605-AF647 (green). The distances were determined by donor-acceptor distance (Equation 5) and radius of QD600, QD630, and streptavidin QD605 (~3.2 nm, ~4.0 nm, and ~3.8 nm, Supplementary Figs. 2 and 16).

Fig. 3. Wireframe DNA origami-based chimeric ssDNA wrapping and biotin-streptavidin conjugation. (a) Schematic and TEM images of Pep-QD assemblies using chimeric ssDNA (30 nt A*) wrapping or biotin-streptavidin conjugation at the inner center and outer edge of Pep wireframe origami objects. The percentage of mono-, di, and tri-valent Pep-QD assemblies were calculated from TEM images (308 and 337 assemblies for Pep with 30 nt A* domain at the inner center and outer edge, respectively; 257 and 265 assemblies for Pep with biotin domain at the inner center and outer edge, respectively). (b) AGE (0.8%) image (four-fold excess Pep) of QD630 alone, Pep-30 nt A*-QD630 (inner center), Pep-30 nt A*-QD630 (outer edge), Pep-30 nt A*(inner center), and Pep-30 nt A* (outer edge). (c) AGE (0.8%) image (four-fold excess QD) of streptavidin QD660 alone, Pep-biotin-streptavidin QD655 (inner center), Pep-biotin-streptavidin QD655 (outer edge), Pep-biotin (inner center), and Pep-biotin (outer edge). Representative TEM images of (d) Pep-30 nt A*-QD630 (inner center), (e) Pep-30 nt A*-QD630 (outer edge), (f) Pep-biotin-streptavidin QD655 (inner center), and (g) Pep-biotin-streptavidin QD655 (outer edge). (Scale bars: 20 nm (a); 50 nm (d), (e), (f), and (g).)

Supplementary Fig. 18. AGE (1.5%) images of a) DNA ladder, scaffold, Pep wireframe origami objects with biotin domain at the inner center and outer edge, and b) DNA ladder, scaffold, Pep wireframe origami objects with 30 nt A* ssDNA wrapping domain at the inner center and outer edge (from left to right). The very bright bands visible below 500 bp are the excess of staple strands visible before purification.

Supplementary Fig. 19. AGE (0.8 %) images of streptavidin QD655 lone, Pep-biotin- streptavidin QD655 assemblies with the biotin at the inner center, Pep-biotin-streptavidin QD655 assemblies with the biotin at the outer edge, Pep-biotin (inner center), and Pep-biotin (outer edge) (from left to right) in 2 h run. Fluorescence images taken by a) gel imaging system and b) digital camera under blue light excitation with ten times Pep alone control.

Supplementary Fig. 20. TEM images of Pep-30 nt A*-QD630 assemblies using ps-backbone based wrapping domain at the inner center at several magnifications. (Scale bars: 50 nm)

Supplementary Fig. 21. TEM images of Pep-30 nt A*-QD630 assemblies using ps-backbone based wrapping domain at the outer edge at several magnifications. (Scale bars: 50 nm)

Supplementary Fig. 22. TEM images of Pep-biotin-streptavidin QD655 assemblies using biotin domain at the inner center at several magnifications. (Red arrow: divalent assemblies. Scale bars: 50 nm)

Supplementary Fig. 23. TEM images of Pep-biotin-streptavidin QD655 assemblies using biotin domain at the outer edge at several magnifications. (Red arrow: divalent assemblies, green arrow: trivalent assemblies. Scale bars: 50 nm)

Supplementary Fig. 24. The yield of Pep-30 nt A*-QD630 assemblies using ps-backbone wrapping domain at the a) inner center and b) outer edge. The yield of Pep-biotin-streptavidin QD655 assemblies using biotin domain at the c) inner center and d) outer edge. Yields were calculated from TEM images (308 assemblies for a, 337 assemblies for b 257 assemblies for c), and 265 assemblies for d)).

- Line 71: The sentence that starts, “In contrast to tile-based self-assembly...” Is unclear. I think it is just worded awkwardly.

Response: The sentence was revised to the following:

“In contrast to tile-based self-assembly approaches that typically suffer from limited yields^{30–32}, wireframe DNA origami typically offers an excellent yield (above 90%) of nearly arbitrary 2D and 3D target DNA-based nanostructures together with site-specific functionalization^{33–35}.”

- Line 78: What is meant by “quantitative fabrication yield”? Do the authors mean the yield is 100%? This is a minor point, but I would not claim 100% yield unless it is fully demonstrated, and typical characterizations, like gel electrophoresis are not sufficient to make the claim of 100% perfectly folded structures.

Response: The sentence was revised to the following:

“In contrast to tile-based self-assembly approaches that typically suffer from limited yields^{30–32}, wireframe DNA origami typically offers an excellent yield (above 90%) of nearly arbitrary 2D and 3D target DNA-based nanostructures together with site-specific functionalization^{33–35}.”

- Figure 2b: I find it difficult to see everything in this plot. I would suggest showing in separate plots the for the QD600, QD630 and AF657 and another for the QD660 and AF750. Also I do not think the “PL” in the Normalized “PL” axis is defined anywhere. Also the colors of letting in the legend is a bit different than the curves making it difficult to map everything together.

Response: We have changed the color in the legend to match to curves color, and define the Photoluminescence (PL) in manuscript and the Figure legend.

“For QD600-AF647 and QD630-AF647 FRET pairs, QD donor photoluminescence (PL) intensities and lifetimes decreased with increasing ps-backbone length from 5 to 30 nt A* due to FRET (Supplementary Figs. 4 and 6).”

“(b) Extinction coefficient (dash-dotted line) and photoluminescence (PL) spectra (solid line) of QD600, QD630, QD660, AF647 and AF750.”

- Lines 196-197: I do not understand the comment, “which may be due to a polydisperse set of conformations of the chimeric ssDNA when wrapped on the QD surface.” To me, a polydisperse set of conformations would have to do with either multiple bands or the thickness of the band, not necessarily a minor shift relative to the control or to other strand lengths.

Response: We apologize for the lack of clarity. We have changed the sentence in the revised manuscript:

“AGE (Supplementary Fig. 9) revealed a single band with minor electrophoretic mobility shifts between QD600 wrapped with 5–40 nt A*, which may be due to a different set of conformations of the chimeric ssDNA when wrapped on the QD surface.”

- Lines 199-201: It does not look to me like the mobility of the QD wrapped with 0 nt A* is similar to the monovalent band of the thiolated-ssDNA-wrapped QD600. The monovalent band of the thiolated-ssDNA-wrapped QD600 runs faster, more like the 5 nt A* or 10 nt A* wrapped QDs. Also, I am confused about claim about the non-specific adsorption. To me the shift of the 0 nt A* relative to the bare QD600 would suggest non-specific adsorption of the 0 nt. And I do not think the fact that the mobilities of the thiolated-ssDNA-wrapped QD600s are similar to any of the A* wrapped QDs would necessarily suggest non-specific surface adsorption of the thiolated QDs. I am not sure if that is what the authors are suggesting. The experiment described later in the paragraph clarifies that there is non-specific adsorption of the AF647 po strand. But the authors should clarify the description of the gel results here.

Response: We apologize for the lack of clarity in our analysis. The mobility of the QD wrapped with 0 nt A* indicate the non-specific adsorption of 0 nt A*. We did not suggest the non-specific surface adsorption of the thiolated QDs. We have revised the sentence in the revised manuscript:

“We observed a shift of QD600 wrapped with 0 nt A*(without any ps-backbone) relative to the bare QD600, indicating some non-specific adsorption of 0 nt A*.”

We also performed the AGE gel experiment of the non-specific adsorption of AF647 po strand (21 nt). After contrast adjustment, AGE images in AF647 channel also showed a small amount of nonspecific DNA adsorption on bare QDs (Supplementary Fig. 10)

Supplementary Fig. 10. a) Schematic of nonspecific DNA adsorption. b) Steady-state PL spectra and c) AGE (0.8%) images of QD600 alone (black), QD 600 and AF647-21 nt po mixture (red), QD600-30 nt A* alone (blue), QD600-30 nt A* (non-complementary po domain) and AF647-21 nt po mixture (green), and AF647-21 nt po alone (violet) (from left to right). d) Contrast adjustment of AGE images in AF647 channel indicate the tiny amount of DNA nonspecific adsorption on bare QD.

- Lines 247-248: How was the mean donor(QD)-acceptor(AF750) distance calculated from TEM? Figure S2 only shows diameter measurements.

Response: The mean donor (QD)-acceptor (AF750) distance was calculated using the radius of QD660 ($\sim 7.1 \pm 1.1$ nm) and the contour length of the polyT spacer (~ 0.68 nm).

- Lines 267-274: It took me awhile to understand what was being shown in these gels. I believe the green signal in in Fig. 3d shows QDs and DNA nanostructures while the yellow only shows QDs. I think it would be helpful to specifically state the emission spectra of the SYBR Safe and the QD600 overlap in the main text description. Also the later description talks about QD600 (Line 273) with respect to Figure 3i, but the 3i shows QD660s (I think that was just a typo). It would also be helpful to label in the gel image or in the figure caption (or both) what colors show what in the two gels in figure 3.

Response: The green signal in Fig. 3d (Fig. 4d in the revised manuscript) shows QDs and DNA nanostructures while the yellow signal is the merge of red signal (QD) and green signal, which still only shows QDs (Supplementary Fig. 32). The description about QD600 (Line 273) was a typo, which we corrected to “QD660”. We have added a statement about how the emission spectra of the SYBR Safe and the QD600 overlap in the main text description as follows, and labeled what color shows what in the two gels in the Figure caption:

“Due to the emission spectra overlaps of the nucleic acid fluorescence stain SYBR Safe and the QD600, these two species were distinguished using two fluorescence channels with different excitation

wavelengths: the QD600 channel combined with SYBR Safe (blue light excitation) and the QD600-only channel (UV light excitation) (Fig. 4d and Supplementary Fig. 32).”

Supplementary Fig. 32. AGE (0.8 %) image of QD600 alone, Tet-QD600 assemblies and Tet wireframe DNA origami objects alone (from left to right). a) Digital images taken by a gel imaging system under UV (QD channel) and blue light (QD and SYBR Safe channel) excitation. b) Image of a blue-light illuminated gel taken using a digital camera. Although the QD600 and SYBR Safe (wireframe DNA origami objects) have similar emission regions, we can distinguish them using the digital camera (iPhone X).

- Line 284: What was the yield of the properly assembled Tet-QD600 assembly? The gel appears to show a leading edge that may only be DNA structure, and it seems there are a few (very few) DNA origami Tets without QDs in figure 3c. My impression is the yield is quite good, but it would be good to quantify.

Response: The yield of Tet-QD600 was quantified by TEM using 358 NPs and up to 91%.

- Lines 309-311: Were TEM images taken of these Tet-QD constructs? Can the authors quantify yield? It appears to be quite good from the gel, but TEM images would be useful to confirm.

Response: The yield of these Tet-QD constructs should be similar to above mentioned constructs, since they are the same assemblies.

- Figure 4C: how were these constructs designed? It seems there is one wrapped chimeric DNA that binds 1-4 AF647. I do not see how that can be done but maintain all of the dyes at ~7nm. Also, how is the 7nm spacing for this construct determined? Is it just based on the length of the duplex? The figure is somewhat confusing also because it shows different color dots as the dyes, are those all AF647? And they are anchored at different positions, but I understood there is only one strand wrapped?

Response: These constructs were designed using the approach from previous literature (*Angew. Chem. Int. Ed.* **56**, 16077–16081 (2017)). As described in Supplementary Methods-Sample preparation for QD-DNA complex constructs, equimolar concentrations of the four types of chimeric DNA were mixed together, and annealed using the reported protocol. As illustrated in Supplementary Fig. 37, the 7 nm spacing was just determined from the length of the duplex (11 bps: ~3.8 nm) and radius of the QD (~3.2 nm), which is “Floppy”.

The reviewer is correct. Those are all AF647 dyes. We have changed the dots to the same color in revised Fig 5a and 5c.

Supplementary Fig. 37. Schematic for preparation of a) DNA complex-QD-based FRET network, b) trimer Type C and c) trimer Type D. The distance between QD and dye was calculated using the duplex length (~ 3.8 nm) and QD radius (~ 3.2 nm) (DNA domains are denoted with lowercase letters, and domains x and x' are complementary to each other).

- Line 332: What is the yield of these? The yields look very good to me, but it would strengthen the results to give a quantitative number, which could be estimated from TEM images, instead of just saying "high fidelity".

Response: The yield of the QD trimer was about 75%, which was calculated by TEM using more than 200 assemblies.

- Lines 336-339: How are these angles designed in the DNA complex-based QD trimers?

Response: These "Floppy" angles were designed using the approach from previous literature (*Angew. Chem. Int. Ed.* **56**, 16077–16081 (2017)). As described in Supplementary Methods-Sample preparation for QD-DNA complex constructs, to fabricate the DNA complex-based QD trimers, DNA complex-QD with valence (II) (Type C and Type D) were incubated with complementary DNA labeled monovalent QD. As illustrated in Supplementary Fig. 37, valence (II) Type C represented the two overhangs for hybridization in close proximity, while Type D represented the two overhangs at a distal position.

Reviewer #3 (Remarks to the Author):

The work by Chen et al reports a use of DNA frames for controlling positions of quantum dots. Although the topic of a broad interest for self-assembly, as for applications in nano-optics, imaging and light manipulation, there are various important aspects that must be addressed.

First general takeaway is that the authors are making claims that go far beyond this work – they aren't the first to use these types of chimeric DNA strands to mono-functionalize quantum dots, nor are they the first to attach materials to DNA origami structures (which they note). The authors seem to be confirming that these bind to overhangs on a DNA origami structure – there are a lot of experiments here, but this seems to be the main point, in addition to claiming that this strategy is more effective than other approaches.

Response: We appreciate the reviewer's concerns. We introduced the history of the chimeric ssDNA wrapping strategy in the Introduction and compared our approach with previous approaches in Supplementary Table 1. We did not intend to claim that we were the first to use these types of chimeric ssDNA to mono-functionalize quantum dots. The main point of this work was the first use of wireframe DNA origami objects and nanoparticle surface modifications to translate nanoscale wireframe DNA origami design strategies into functional hybrid nanoparticle materials. And to demonstrate that the chimeric ssDNA wrapping approach can be applied to the overhang of a DNA origami structure, which can not only realize mono-functionalized QD-DNA origami assemblies with 3D spatially addressable valence, but also minimize the distance between the QD and wireframe DNA origami object.

Supplementary Table 1. Comparison of various DNA-based strategies to engineer valences on QDs.

Reference [citation number in main MS]	Method	Controllable valence number	Controllable valence position
Nat. Nanotechnol. 2011[16]	one pot synthesis (ps+po backbone)	1-5	no
Nat. Methods 2013[17]	ssDNA wrapping (ps+po backbone)	1	N/A
Angew. Chem. 2017[18]	ssDNA wrapping (programmable DNA complexes)	1-4	no
This work	ssDNA wrapping and folding (QD-wireframe DNA origami objects assemblies)	1 to >5 (limited only by the number of overhangs on the wireframe DNA origami)	yes (valence-geocoded)

Second takeaway – given the conclusion of this work that the DNA wrapping path towards mono-functionalization is better than more traditional methods of DNA attachment to quantum dots, both because of the valence and because of the bulkiness of certain functional groups such as biotin-streptavidin. The paper even goes through measurements to get these actual values for spacing, distance, some function for a FRET dye network. However, none of this compared to any of the systems the authors are claiming need to be improved upon. There is need for control experiments and comparisons, otherwise these claims are not supported. The authors use of a thiolated DNA sequence earlier in the paper to show that the thiolated

sequence leads to multiple attached DNA strands – they felt they should show a control here with a more traditional ssDNA (though don't understand the need for the thiol group as it doesn't appear to be used). However, this experimental "control" group is missing everywhere else in the paper.

Response. In our revised manuscript we designed a series of experiments to compare our approach with thiolated-ssDNA and biotin-streptavidin. Compared with monovalent QD-dye FRET, FRET efficiency of QD-thiol-dye FRET pairs calculated from steady-state measurement increased from $69 \pm 3\%$ to $81 \pm 2\%$, from $59 \pm 2\%$ to $74 \pm 7\%$, and from less than 5% to $40 \pm 4\%$ for QD600-thiol-AF647, QD630-thiol-AF647, and QD660-thiol-AF750, respectively, indicating the presence of multiple dye acceptors due to multivalent binding between QD and 3'-thiolated-ssDNA (Supplementary Fig. 4). And the FRET efficiency of QD660-thiol-AF750 FRET pairs increased more significantly due to the larger surface area that can potentially bind more ssDNAs.

To further illustrate the advantage of the ssDNA wrapping strategy, commercial streptavidin QD605 was selected to fabricate the QD-streptavidin-biotin-AF647 FRET pairs (Supplementary Figs. 16 and 17). Considering the $72 \pm 4\%$ of FRET efficiency, 7.0 nm of Förster distance and 9.8 nm of donor-acceptor distance, the number of AF647 per streptavidin QD605 was calculated as 19 ± 4 using Equation 5 (Methods), which is consistent with the result calculated from three available biotin-binding sites after conjugation to the streptavidin QD605 and there are 5-7 streptavidins (provided by the commercial vendors) bound on each streptavidin QD605 on average (Supplementary Fig. 17). This result indicates several drawbacks of streptavidin QDs for precise nanofabrication, including uncontrollable DNA binding and the larger donor-acceptor distance (~ 10 nm) due to the multivalent and bulky streptavidin-biotin conjugation.

Moreover, a series of experiments were designed to compare wireframe DNA origami-based chimeric ssDNA wrapping and biotin-streptavidin conjugation. Briefly, Pentagonal pyramid (Pep) wireframe origami objects with an overhang containing ps-backbone (30 nt A*) at the inner center or outer edge were used to fabricate Pep-30 nt A*-QD630 assemblies. In contrast, Pep with a biotin-modified overhang at the inner center or outer edge were used to fabricate Pep-biotin-streptavidin-QD655 assemblies (Fig. 3a). AGE gel shift (Figs. 3b, 3c and Supplementary Figs. 18 and 19) and negative-stain TEM imaging (Figs. 3a, and 3d-3g, Supplementary Figs. 20-23) of the Pep-QD assemblies validated the assembly of the target DNA origami objects and different stoichiometric ratios of Pep per QD using chimeric ssDNA wrapping and streptavidin-biotin conjugation. By using chimeric ssDNA wrapping, each QD can only bind with a single Pep wireframe DNA origami object with an overhang at inner center or outer edge. However, in the case of biotin-streptavidin conjugation, a streptavidin QD that contains multiple valences (active streptavidins) can bind one (monovalent) or two (divalent) Pep with a biotin-modified overhang at inner center, and up to three (trivalent) Pep with a biotin modified overhang at outer edge. Although the loading yields (Pep-QD/Pep) of Pep-30 nt A*-QD630 (inner center: 85%; outer edge: 87%) and Pep-biotin-streptavidin QD655 (inner center: 90%; outer edge: 86%) were similar, the yield of the correct assemblies (monovalent Pep-QD/Pep) was nearly equal to the loading yield for the chimeric ssDNA wrapping strategy, but only 68% for biotin-streptavidin conjugation at inner center, and 61% for biotin-streptavidin conjugation at outer edge (Supplementary Fig. 24). This result further demonstrated that the monovalent ssDNA wrapping was crucial for efficient and correct QD-DNA origami-based nanofabrication.

We have discussed the above-mentioned results in the revised manuscript on pages 13-16.

Supplementary Fig. 4. Photoluminescence (PL) spectra of a) QD600-AF647, b) QD630-AF647, and c) QD660-AF750 FRET pairs. d) FRET efficiency calculated from each FRET system using various ps backbone length and 3'-thiolated ssDNA (red: 5A*, blue: 10A*, green: 30A*, brown: 50A*, and cyan: 3'-thiolated-ssDNA).

Supplementary Fig. 16. TEM images (bottom) and size distribution analysis (top) for commercial a) streptavidin QD605 and b) streptavidin QD655. Particle diameters were calculated from 567 NPs (a) and 515 NPs (b), respectively. Red curves are Gaussian fit to the measured particle diameter distribution. Indicated particle diameters for each histogram are expressed as mean \pm standard deviation.

Supplementary Fig. 17. a) Schematic of streptavidin QD605 binding with biotin-modified AF647- labelled DNA duplex. b) PL spectra of streptavidin QD605-AF 647 FRET pairs. c) FRET efficiencies as a function of acceptors calculated theoretically (black curves, Eq. 5), and from QD emission intensities (Supplementary Fig. 17b). The donor-acceptor distances of 9.8 nm were determined by radius of streptavidin QD605 (~ 3.8 nm, Supplementary Fig. 16a), streptavidin-biotin conjugation (~ 5 nm), and polyT spacer (~ 1 nm). d) Schematic of minimum distance calculation of streptavidin QD-dye FRET system (left) and monovalent wrapping QD-dye FRET system (right).

Fig. 3. Wireframe DNA origami-based chimeric ssDNA wrapping and biotin-streptavidin conjugation. (a) Schematic and TEM images of Pep-QD assemblies using chimeric ssDNA (30 nt A*) wrapping or biotin-streptavidin conjugation at the inner center and outer edge of Pep wireframe origami objects. The percentage of mono-, di, and tri-valent Pep-QD assemblies were calculated from TEM images (308 and 337 assemblies for Pep with 30 nt A* domain at the inner center and outer edge, respectively; 257 and 265 assemblies for Pep with biotin domain at the inner center and outer edge, respectively). (b) AGE (0.8%) image (four-fold excess Pep) of QD630 alone, Pep-30 nt A*-QD630 (inner center), Pep-30 nt A*-QD630 (outer edge), Pep-30 nt A*(inner center), and Pep-30 nt A* (outer edge). (c) AGE (0.8%) image (four-fold excess QD) of streptavidin QD660 alone, Pep-biotin-streptavidin QD655 (inner center), Pep-biotin-streptavidin QD655 (outer edge), Pep-biotin (inner center), and Pep-biotin (outer edge). Representative TEM images of (d) Pep-30 nt A*-QD630 (inner center), (e) Pep-30 nt A*-QD630 (outer edge), (f) Pep-biotin-streptavidin QD655 (inner center), and (g) Pep-biotin-streptavidin QD655 (outer edge). (Scale bars: 20 nm (a); 50 nm (d), (e), (f), and (g).)

As a schematic diagram Figure 1 does a good job outlining the method that the authors used in this study. As for the results shown in Figure 2, I either cannot see major differences to the use of chimeric DNA to previous results by other groups or the authors did not convey those points clearly enough in the writing. The authors of this study even cite previous work using this method, but use Figure 2 to essentially reaffirm this method, unrelated to the DNA origami design at all.

Response: We used the previous method to prepare the DNA wrapped QD. The previous result by another group studied the impact of ps-backbone length on wrapping efficiency, which did not systematically consider the impact of QD size and hybridization efficiency of the po domain. Jun et al. concluded that the ssDNA containing 70 mers ps-backbone length can achieve monovalent wrapping (*Nat. Methods* 10, 1203–1205 (2013)), while Fan et al. concluded ssDNA containing 30 mers ps-backbone length got the highest monovalent wrapping efficiency (*Angew. Chem. Int. Ed.* 56, 16077–16081 (2017)). Here, we systematically investigated the impact of ps-backbone length and QD size on valence control and hybridization yield. We found that QD size is crucial for the selection of ps-backbone length, larger sized QDs (14 nm) having more active anchor sites for DNA wrapping, increasing the probability of wrapping more than one chimeric ssDNA. Moreover, the shorter ps-backbone length leads to a conformational change in the ssDNA and reduces the hybridization efficiency of the po domain, supported by MD simulations. Then the overhangs of DNA origami were designed with different ps-backbone lengths for differently sized QD wrapping. This design was crucial for the overhang in the inner center of the semi-surrounded DNA origami structure and the overhang at the outer edge of the DNA origami structure. As shown in Fig 3, by using a 30 nt A* chimeric ssDNA wrapping, QD630 could only bind with a single Pep wireframe DNA origami object with an overhang at the inner center or outer edge due to the monovalent wrapping. QD660 can bind two or three Pep wireframe DNA origami objects using a 30 nt A* overhang at the outer edge (Additional Fig. 1), and can bind two Pep wireframe DNA origami objects even using a 50 nt A* overhang at the inner center (Supplementary Fig. 36), which is consistent with the result from Fig. 2e. Similarly, in the case of the biotin-streptavidin conjugation, streptavidin QDs that contain multiple valences (active streptavidins) can bind one (monovalent) or two (divalent) Pep with a biotin-modified overhang at the inner center, and up to three (trivalent) Pep with a biotin modified overhang at the outer edge.

Additional Fig. 1. Representative TEM images of Pep-30 nt A*-QD660 with the chimeric ssDNA wrapping domain at outer edge.

The paper seems unclear what its main points are. Much of the beginning half of the paper is dedicated to work that has, in large part, been previously published. All of this is supposedly done to build a picture of control for binding to origami structure to enable functionality not truly achievable with other methods of functionalizing quantum dots. Yet, while Figure 3 and 4 do get into binding to origami, there is never a comparison of structural yields for binding or functionality of the quantum dots in comparison to other

systems. The discussion then seems to turn to the rigidity of the origami structure – which again, is already known, and has been shown many times before. Effectively, the story seems to change in the middle.

Response: We appreciate that experiments bridging the ssDNA wrapping and QD-origami assemblies were missing in our previously submitted manuscript. To emphasize the relationship between ssDNA wrapping and fabrication of QD-origami assemblies, and to compare our approach to other systems, we designed and performed a series of experiments to demonstrate the ps-backbone length on chimeric ssDNA, QD size, and overhang positions of wireframe DNA origami objects were crucial to achieve monovalent QD-DNA origami assemblies. As shown in Fig 3, by using a 30 nt A* chimeric ssDNA wrapping, QD630 can only bind with a single Pep wireframe DNA origami object with overhang at the inner center or outer edge due to the monovalent wrapping. QD660 can bind two Pep wireframe DNA origami objects even using a 50 nt A* overhang at the inner center (Supplementary Fig. 36), which is consistent with results from Fig. 2e. Similarly, in the case of biotin-streptavidin conjugation, streptavidin QDs that contain multiple valences (active streptavidins) can bind one (monovalent) or two (divalent) Pep with a biotin-modified overhang at the inner center, and up to three (trivalent) Pep with a biotin modified overhang at the outer edge. Although the loading yields (Pep-QD/Pep) of Pep-30 nt A*-QD630 (inner center: 85%; outer edge: 87%) and Pep-biotin-streptavidin QD655 (inner center: 90%; outer edge: 86%) were similar, the yield of the correct assembly (monovalent Pep-QD/Pep) was nearly equal to the loading yield for the chimeric ssDNA wrapping strategy, but only 68% for biotin-streptavidin conjugation at the inner center, and 61% for biotin-streptavidin conjugation at the outer edge (Supplementary Fig. 24). This result further demonstrated that the monovalent ssDNA wrapping was crucial for the efficient and correct QD-DNA origami-based nanofabrication.

In Figure 3, the particles are semi-surrounded by the structure they are binding to, thus negating the argument that mono-functionalization leads to reduced aggregation of structure because otherwise the particles could bind other structures if they have multiple binding sequences. However, the specific structures the authors tested would cloud this conclusion in the first place – it is not clear why a multi-valent quantum dot would give any different results in the structures shown in Figure 3, given that the structure most effectively binds one quantum dot, and then sterically hinders the quantum dot from binding to other structures.

Response: The Tet structure with overhang at the inner site can effectively bind one quantum dot, and then sterically hinders the quantum dot from binding to other structures. However, in the case of the Pep structure, and origami structure with overhang at the outer edge (e.g., fabrication of QD-based colloidal molecules (Fig. 6)), the QD could still bind the other structures if they had multiple binding sequences (Additional Fig. 1). We tested the impact of multiple binding sites and overhang positions in Fig. 3.

Figure 3 focuses on the localization of QDs to the DNA origami. The use of gel and TEM here does a good job to show that the QDs become fixed to their target site, however this figure would be a good place to show statistics. The authors show in Supplementary Fig. 24 that to achieve a 70% yield of correctly placed monovalent QDs, they needed to use 4:1 molar excess, otherwise a 1:1 would result in a 52% loading efficiency. So, I am still not seeing the real need for chimeric DNA.

Response: We have now shown the statistics in Figs. 3 and 4 in revised manuscript.

The monovalent yield of Pep-50 nt A*-QD660 was relatively lower because the Pep with 50 nt A* ssDNA could lead to more than one wireframe DNA origami object wrapping (20%) according to the results of our previous ssDNA wrapping hypothesis section (Fig. 2e). For QD630 that showed monovalent wrapping using 30 nt A* ssDNA, the yield of the correct assemblies was nearly equal to the loading yield.

We noticed that the 4:1 molar excess of QDs can help to increase the monovalent yield of Pep-50 nt A*-QD660; however, a high proportion of di- and tri-valent Pep-QD assemblies could still form even when the Pep-biotin were incubated with four-fold excess streptavidin QD655 (Figs 3c, 3f and 3g). This difference could be explained by the high-affinity biotin-streptavidin interaction ($K_d = 10^{-14}\text{M}$), the activated streptavidin binding sites can still bind other Pep wireframe DNA objects, even after formation of Pep-QD assemblies. However, in the case of Pep-50 nt A* QD660 (inner center), the excess QD could efficiently decrease the probability of QD binding two ps-backbone overhangs simultaneously, then the Pep structure and overhang wrapping on QD sterically hinders the QD from binding to other structures due to the relatively weaker interaction. This result further demonstrated that the monovalent ssDNA wrapping is crucial for efficient and correct QD-DNA origami-based nanofabrication.

Figure 4, as stated previously, seems to lose the story. The control used for a comparison here is a floppy DNA linking structure, as opposed to the structural DNA origami system. The point of the paper wasn't that DNA origami can organize materials more rigidly and effectively than a floppy DNA structure, it was the quantum dot functionalization technique yielded differences in quantum dot function and organizational yield when compared to other techniques. Neither are shown.

Response: Thank you for pointing out this issue. The main point of this work is the use of wireframe DNA origami objects and nanoparticle surface modifications to translate nanoscale DNA origami design strategies into functional hybrid nanoparticle materials, such as energy-transfer circuits and colloidal molecules. And the chimeric ssDNA wrapping approach can be applied to the overhang of a DNA origami structure, which can not only achieve the mono-functionalized QD-DNA origami assemblies, but also minimize the distance between the QD and wireframe DNA origami object. The mono-functionalized QD-DNA origami assemblies was crucial for the fabrication of QD-based colloidal molecules since the multiple binding could otherwise produce a cluster of QD-DNA origami assemblies. In order to emphasize the importance of minimization of distance between the QD and wireframe DNA origami object, QD600-AF647-AF750 concentric multi-step FRET networks were fabricated using Pep wireframe DNA origami objects in the revised manuscript. In contrast, the same multi-step FRET networks were fabricated using streptavidin QD605 instead of QD600 (Fig. 5e and Supplementary Figs. 41 and 42). Unlike the Tet wireframe DNA origami object, the Pep wireframe DNA origami object allowed us to fabricate asymmetric QD-dye assemblies. The AF647 was incorporated into the Pep during the DNA wireframe origami folding, and the assembly fidelities were analyzed using AGE with SYBR Safe and AF647 detection channels (Supplementary Fig. 43). The overhangs of a Pep wireframe DNA origami object were designed to hybridize with AF750 dyes on a close (~ 7.1 nm), median (~ 7.8 nm) or far (~ 12.2 nm) distance to AF647 but similar distance to QD (9-10 nm) (Fig. 5e and Supplementary Figs. 41 and 44). Experimental results from the QD600-AF647-AF750 multi-step FRET network were in agreement with theoretical calculations (Figs 5f and 5g, and Supplementary Fig. 45), whereas results from streptavidin QD605 exhibited much lower energy-transfer efficiency between initial donor and relay due to the bulky streptavidin tags, and failed to perform the multi-step FRET process (Supplementary Figs. 42 and 45).

We would like to retain the comparison between floppy DNA linking structure and structural DNA origami since it is one of the advantages of our approach.

My general thought with this paper is that they initially convey to the reader that there are major issues with the more conventional methods of QD localization (biotin-streptavidin) when it comes to FRET probes or networks and the possibility of the same QD being attached to two separate sites. However, there is no control experiments, either in the assembly of the QDs to the origami or measuring the FRET efficiency where the biotin-streptavidin method was used. By claiming that this method resolves these issues but by not providing any qualitative or quantitative benchmarks, the reader is left unclear as to how much this method improves results compared to these others.

I don't see how the paper could be published without additional experiments and clarifying the claims.

Response: Thank you for pointing out this issue. We designed and performed the suggested control experiments, including self-assembly of the QD to the origami and multi-step FRET networks using biotin-streptavidin in the revised manuscript. As shown in Fig 3, the streptavidin QD that contained multiple valences (active streptavidins) could bind one (monovalent) or two (divalent) Pep with a biotin-modified overhang at the inner center, and up to three (trivalent) Pep with a biotin modified overhang at the outer edge. Although the loading yields (Pep-QD/Pep) of Pep-30 nt A*-QD630 (inner center: 85%; outer edge: 87%) and Pep-biotin-streptavidin QD655 (inner center: 90%; outer edge: 86%) were similar, the yield of correct assemblies (monovalent Pep-QD/Pep) was nearly equal to loading yield for the chimeric ssDNA wrapping strategy, but only 68% for biotin-streptavidin conjugation at inner center, and 61% for biotin-streptavidin conjugation at outer edge (Supplementary Fig. 24). This result further demonstrated that the monovalent ssDNA wrapping was crucial for efficient and correct QD-DNA origami-based nanofabrication. Figs. 5e-5g and Supplementary Fig. 42 and 45 show the fabrication of multi-step FRET networks using chimeric ssDNA wrapping and biotin-streptavidin conjugation. Experimental results from the Pep-30 nt A*-QD600-AF647-AF750 based-multi-step FRET networks were in agreement with theoretical calculations (Figs 5f and 5g, and Supplementary Fig. 45), whereas results from Pep-biotin-streptavidin QD605-AF647-AF750 FRET networks exhibited considerably lower energy-transfer efficiency between initial donor and relay due to the bulky streptavidin tags, and failed to realize the multi-step FRET process. These results further demonstrated that our approach can minimize the distance between additional functionalities and the QD surface at 2.8 nm for Pep-30 nt A*-QD600 assemblies, while the distance was up to ~7.4 nm for Pep-biotin-streptavidin-QD605 assemblies.

Supplementary Fig. 42. a) Schematic of Pep-30 nt A* QD600-AF647 (left), Pep-30 nt A* QD630-AF647 (middle), and Pep-biotin-streptavidin QD605-AF647 (right). Representative PL spectra of b) Pep-30 nt A* QD600-AF647, c) Pep-30 nt A* QD630-AF647, and d) Pep-biotin-streptavidin QD605-AF647. e) AF647 to QD surface distances in Pep-30 nt A* QD600-AF647 (red), Pep-30 nt A* QD630-AF647 (blue), and Pep-biotin-streptavidin QD605-AF647 (green). The distances were determined by donor-acceptor distance (Equation 5) and radius of QD600, QD630, and streptavidin QD605 (~3.2 nm, ~4.0 nm, and ~3.8 nm, Supplementary Figs. 2 and 16).

Supplementary Fig. 45. a) Schematic of Pep-30 nt A*-QD600-AF647-AF750 and Pep-biotin-streptavidin QD605-AF647-AF750-based multi-step FRET networks. Representative PL spectra (top) and zoomed area of b) Pep-30 nt A*-QD600-AF647-AF750-based multi-step FRET networks and c) Pep-biotin-streptavidin QD605-AF647-AF750-based multi-step FRET networks.

Some other comments:

A* has a shifting definition through the paper, sometimes referring to just Ps and sometimes referring to Ps + Po. For example, lines 152-154 say the A* contains the Po sequence, but otherwise does not seem like it does.

Response: Thank you for pointing out this issue. We have unified our definition in the revised manuscript. A* just refers to the ps-backbone. And the sentence has been revised as follow:

“All A* tracts also combined with a ssDNA sequence with po-backbone available for hybridization (Fig. 2a).”

The “floppy” DNA complex-QD-dye system did not seem that much worse than the rigid structure – can the authors define what they are defining as significant? In multiple setups, the FRET differences don’t appear to be statistically significant. Also, “Floppy” structures in Figure 4 - what is this structure? Is there a different design? What is the DNA complex-QD-dye system – Figure 4l isn’t enough to describe this. There isn’t sufficient explanation here how are formed in the first place. Are these formed through the mechanism in Reference 18?

Response: We respectfully beg to differ with the reviewer. In the case of multi-acceptors, the FRET differences don’t appear to be statistically significant since the floppy structure provided a similar average distance to the rigid structure when they were identical acceptors. However, there was a significant difference when there were only one or two acceptors (22.9 % vs 32.0 %, 36.4 % vs 42.4% (* $P < 0.05$)). This difference would seriously affect the fabrication of concentric multi-step FRET networks using various

dye acceptors (Figs. 5e-5g). These changes can result in significant deviations from the expected spatial position of QDs and dyes on the wireframe DNA origami, as shown by the FRET efficiency data.

“Floppy” structures were shown in Supplementary Fig. 37. These constructs were designed using the mechanism from previous literature (*Angew. Chem. Int. Ed.* **56**, 16077–16081 (2017)). As described in Supplementary Methods-Sample preparation for QD-DNA complex constructs, equimolar concentrations of the four types of chimeric DNA were mixed together, and annealed using the reported protocol. To prepare QD-DNA complex constructs with valences of (I)–(IV), the prepared DNA complex with valences (I)–(IV) were incubated with QDs overnight. To fabricate DNA complex-QD-dye FRET pairs, DNA complex-QD with valence (I)–(IV) were incubated with complementary DNA labeled with AF647. To fabricate the DNA complex-based QD trimers, DNA complex-QD with valence (II) (Type C and Type D) were incubated with complementary DNA labeled monovalent QD.

Fig. 5. Valence-geocoded QD-based energy transfer circuits. (a) Schematic of the Tet-QD-based FRET network and the overhang design (the distance for each de-excitation pathway is ~ 7 nm). (b) AGE (1.5%) image for 1-4 AF647-labeled Tet wireframe DNA origami objects. (Green: SYBR safe channel, Red: AF647 channel). (c) Schematic of the DNA complex-QD-based FRET network. (d) FRET efficiencies as a function of acceptors calculated from the Tet-QD-dye system, the DNA complex-QD-dye system, and Förster theory (Eq. 5). P -values are from Student’s t -test ($*P < 0.05$); all P -values are provided in Supplementary Table 3. (e) Schematics of three types of Pep-QD-based multi-step FRET networks and predictions of distances between initial donor (QD), relay (AF647), and acceptor (AF750). (f) QD quenching efficiencies and (g) AF647 quenching efficiencies calculated from the Pep-QD based multi-step FRET networks (darker bar), and Förster theory (lighter bar) (Eqs. 6-13). Error bars in Figs. 5d, 5f, and 5g represent standard deviations of the mean ($n = 3$ replicates per group).

Supplementary Fig. 37. Schematic for preparation of a) DNA complex-QD-based FRET network, b) trimer Type C and c) trimer Type D. The distance between QD and dye was calculated using the duplex length (~ 3.8 nm) and QD radius (~ 3.2 nm) (DNA domains are denoted with lowercase letters, and domains x and x' are complementary to each other).

Reviewer #4 (Remarks to the Author):

The manuscript "Nanoscale 3D Spatial Addressing and Valence Control of Quantum Dots using Wireframe DNA Origami" describes a framework for attaching quantum dots (QDs) to addressible sites on DNA origami. The use of origami provides precise control over the positioning of QDs and ordinary fluorophores, which has obvious advantages for sensing applications, but in principle could have impacts on technologies ranging from quantum computing to light bulbs. The work is timely and, overall, well presented, though I have a few concerns about the simulation portion of the study. Hence, I recommend the manuscript for publication provided the authors address the specific concerns raised below.

Response: We thank the reviewer for the very positive feedback. We have addressed their concerns point-by-point below.

Specific Concerns:

1. It's not immediately clear what the consistent-valence force field used for the simulations actually is. The only reference listed is for the commercial BIOVIA package; did the BIOVIA team create this force field? The authors included the force field parameters in the SI, but it's not clear how they would map onto the ssDNA atoms, at least without access to the BIOVIA package. If a reference for the force field is unavailable, it would be helpful to cite other papers using the force field to model similar systems.

Response: We appreciate the reviewer for this comment. The BIOVIA team did not develop this consistent-valence force field (CVFF), they only coded the parameters into the software. Detailed information about CVFF can be found on this website:

<http://www.uoxray.uoregon.edu/local/manuals/biosym/discovery/General/Forcefields/CVFF.html>

CVFF is a standard MD simulation force field, in which the bond, angle, and dihedral interactions are all modelled using harmonic potentials.

Our recent paper used the same CVFF force field to simulate ssDNA binding QDs. [ref.: <https://doi.org/10.1021/acsnano.2c01178>]

CVFF has also been used to model different types of polymers, such as pentiptycene-based poly(o-hydroxyimide) copolymers [ref.: <https://doi.org/10.1021/acsapm.1c00128>] and polymer electrolytes [ref.: <https://dx.doi.org/10.1021/acs.jpcc.9b11689>].

2. The simulations should be longer. A five nanosecond simulation isn't long by contemporary standards even when explicit solvent is used (AMBER, for example, can simulate hundreds of nanoseconds/day). Single-stranded DNA is a very flexible molecule, so quantities like the radius of gyration or the persistence length should be averaged over a large ensemble of conformations. The authors would ideally extend the simulations and provide evidence that averages are converged.

Response: For our model, simulations of a 6 nm QD ran at ~2.5ns/day using 24 CPU cores on one CPU node, and the 14 nm QD ran at ~0.95 ns/day using 1 GPU card on one GPU node. To address the concern of the reviewer, we have extended our simulation time to 10 ns and with 5 trajectories, thereby providing a better representation of the ensemble. Supplementary Fig. 12 indicates that our simulations have indeed converged.

Supplementary Fig. 12. a) The distance of the 5'-end of the po-backbone of (5 nt A*(left), 10 nt A*(middle), and 30 nt A*(right)) + po (23 nt) to the QD surface observed in five 10 ns simulated trajectories. b) Radius of gyration (R_g) analysis of the po domain along the same trajectories. (Five independent trajectories (thin lines) were propagated from an equilibrated structure using 5 different random seeds for the stochastic solvent response, and the time-dependent average (thick line) is also shown here).

3. Supporting animations of the MD simulations may give a sense of the fluctuations and dynamics of the DNA.

Response: We appreciate this suggestion from the reviewer. We have uploaded MD simulations animations in the SI materials.

REVIEWERS' COMMENTS

Reviewer #1 (Remarks to the Author):

The authors addressed most of the concerns of the reviewers and performed a set of entirely new experiments, where they investigate the FRET (or rather quenching) efficiency of various QD / fluorophore arrangements on their wireframe DNA structures. I believe the manuscript now passes the threshold for being published in Nature Communications.

Reviewer #3 (Remarks to the Author):

The authors have provided detailed reply to my comments and done an excellent job in performing required verifications. While I do not find the study to be a breakthrough, the presented approach can be useful. I support the publication.

Reviewer #4 (Remarks to the Author):

The authors have addressed my concerns to my satisfaction and I recommend the manuscript for publication.

REVIEWERS' COMMENTS

Reviewer #1 (Remarks to the Author):

The authors addressed most of the concerns of the reviewers and performed a set of entirely new experiments, where they investigate the FRET (or rather quenching) efficiency of various QD / fluorophore arrangements on their wireframe DNA structures. I believe the manuscript now passes the threshold for being published in Nature Communications.

Answer: Thank you for your positive response to our work.

Reviewer #3 (Remarks to the Author):

The authors have provided detailed reply to my comments and done an excellent job in performing required verifications. While I do not find the study to be a breakthrough, the presented approach can be useful. I support the publication.

Answer: Thank you for your positive response to our revision.

Reviewer #4 (Remarks to the Author):

The authors have addressed my concerns to my satisfaction and I recommend the manuscript for publication.

Answer: Thank you for your kind comments.

Draft Only